# Identification of transmissible proteotoxic oligomer-like fibrils that expand conformational diversity of amyloid assemblies

Phuong Trang Nguyen[1,2], Ximena Zottig[1,2], Mathew Sebastiao[1,2], Alexandre A. Arnold[1,2], Isabelle Marcotte[1,2] & Steve Bourgault [1,2✉]

Protein misfolding and amyloid deposition are associated with numerous diseases. The detailed characterization of the proteospecies mediating cell death remains elusive owing to the (supra)structural polymorphism and transient nature of the assemblies populating the amyloid pathway. Here we describe the identification of toxic amyloid fibrils with oligomer-like characteristics, which were assembled from an islet amyloid polypeptide (IAPP) derivative containing an Asn-to-Gln substitution (N21Q). While N21Q filaments share structural properties with cytocompatible fibrils, including the 4.7 Å inter-strand distance and β-sheet-rich conformation, they concurrently display characteristics of oligomers, such as low thioflavin-T binding, high surface hydrophobicity and recognition by the A11 antibody, leading to high potency to disrupt membranes and cause cellular dysfunction. The toxic oligomer-like conformation of N21Q fibrils, which is preserved upon elongation, is transmissible to naïve IAPP. These stable fibrils expanding the conformational diversity of amyloid assemblies represent an opportunity to elucidate the structural basis of amyloid disorders.

[1] Department of Chemistry, Université du Québec à Montréal, Montreal, QC, Canada. [2] Quebec Network for Research on Protein Function, Engineering, and Applications, PROTEO, Quebec, QC, Canada. ✉email: bourgault.steve@uqam.ca

Misfolding and aggregation of proteins into ordered cross-β amyloid assemblies are associated with over fifty human diseases, including the Alzheimer's disease (AD), type II diabetes mellitus and systemic amyloidosis[1]. Amyloid fibril formation and tissue deposition are initiated by the self-recognition of (partially) unfolded proteins and implicate a infinite array of off- and on-pathway prefibrillar intermediates[2]. Compelling experimental and clinical evidence have indicated that the most cytotoxic proteospecies of the amyloid cascade are the oligomeric intermediates and that fibrils are generally non-toxic[1,3]. For instance, the amyloid-β (Aβ) peptide impairs synaptic plasticity in absence of fibrils, through the formation of oligomeric intermediates[4], while the influx of $Ca^{2+}$ across neuronal membranes correlates closely with binding of Aβ oligomers to the plasma membrane[5]. Cytotoxicity, including persistent ability to disrupt cellular membranes, has been reported for oligomers assembled from numerous amyloidogenic proteins, including the islet amyloid polypeptide (IAPP)[6,7], α-synuclein[8,9] and transthyretin[10,11]. Strikingly, cytotoxic oligomers have been generated from proteins that do not spontaneously self-assemble and that are not associated with amyloid disorders[12,13], suggesting that prefibrillar aggregates share physicochemical and/or structural properties mediating toxicity.

Although these studies have supported the oligomer hypothesis, i.e. that soluble aggregates are the main causative agents of cell death, prefibrillar assemblies are transient and heterogeneous, leading to major challenges for elucidating their structures and mechanisms of toxicity. Besides, thorough studies have periodically reported the cytotoxicity of well-defined fibrils[14], including those assembled from IAPP[15], huntington protein[16], β2-microglobulin[17,18] and α-synuclein[19]. Discrepancies among the relationships between toxicity and (supra)molecular architecture of amyloid proteospecies originate from their high structural diversity at the atomistic and mesoscopic levels and/or from their dynamic nature. A given protein can misfold and aggregate into multiple populations of oligomers and fibrils, and each of these quaternary conformers can display divergent cytotoxicity. For instance, Aβ peptide oligomerization can lead either to stable non-toxic dimers/trimers[20], or to pentamers that are very toxic to neuronal cells[21]. By varying self-assembly conditions, two populations of stabilized α-synuclein oligomers with similar sizes and morphologies were obtained; one type being benign to cells and the other population avidly perforating lipid bilayers[8,22]. Similarly, under different temperatures, the huntington protein with extended polyglutamine misfolds into conformational distinct fibrils with divergent neurotoxicity[16]. While some supramolecular organizations mediating toxicity have been proposed, including the β-barrel structure of Aβ peptide oligomers[23], the annular conformation of α-synuclein oligomers[8] and the pairs of β-sheets mated by a dry interface for IAPP fibrils[15], it has also been proposed that cytotoxicity could be independent of a given structure[24]. Thus, the elucidation of the toxic conformation(s) of the amyloid cascade is still the matter of active debates owing to the large ensemble of on- and off-pathway transient species assembled along the aggregation pathway.

Studies have exploited site-directed mutagenesis to define the chemical determinants of self-assembly and to elucidate the conformational rearrangements triggering oligomerization, nucleation and/or fibril elongation. Structure-self-assembly relationship studies often lead to the identification of residues modulating aggregation and of distinct assemblies with unique underlying mechanisms of toxicity[25,26]. Using such an approach, we recently reported that N21 of IAPP behaves as a molecular hinge controlling conformational conversion and toxicity toward pancreatic β-cells[27]. IAPP is a 37-residue peptide hormone whose deposition as insoluble amyloids in the pancreatic islets is associated with type II diabetes[28]. In the present study, we report that the apparently trivial Asn-to-Gln substitution at position 21 of IAPP, i.e. the addition of a single methylene group into a 4 kDa peptide, led to the formation of fibrils with high cytotoxicity. These defined prototypical fibrils displayed oligomer-like properties, such as exposition of hydrophobic clusters, recognition by the conformational A11 antibody, and high capacity to perturb plasma membrane. These stable oligomer-like fibrils reveal that the oligomer conformation can be transmissible and represent a unique opportunity to elucidate the structure of toxic amyloid proteospecies.

## Results

**Shifting the toxicity of amyloid fibrils by an N-to-Q substitution.** IAPP is known for its propensity to self-assemble into cross-β amyloid structures, which are associated with the pathogenesis of type II diabetes. As reported by solid-state NMR, each monomeric unit of the amyloid fibrils are composed of two β-strands connected by a disordered loop involving positions H18 to L27[29]. Recent cryo-EM studies have revealed conformational polymorphism of IAPP fibrils, with monomers composed of three β-sheets of variable lengths connected by disordered loops[30–32]. Notwithstanding the models of IAPP fibrils, residue N21, which is critical for self-assembly and toxicity[27,33,34], is located in a disordered loop joining β-strands and its side chain projects outward into protofilament interface, likely participating in protofilament packaging (Fig. 1a). As reported for other amyloidogenic proteins, fibrils assembled from IAPP are poorly cytotoxic, whereas oligomers and prefibrillar species damage cells[7,35]. Strikingly, the substitution of Asn at position 21 by a Gln (N21Q), which corresponds to displacement of the amide group away from the peptide backbone by approximatively 1.5 Å, led to the formation of fibrils that are highly toxic to β-pancreatic cells (Fig. 1). Fibrils were prepared by incubating the monomerized peptide in 20 mM Tris-HCl, pH 7.4 for 48 h at 150 µM under quiescent conditions at room temperature. Under these conditions, N21Q derivative self-assembled into prototypical twisted filaments (Fig. 1b), showing a concentration-dependent cytotoxicity. Viability measured by the metabolic activity of pancreatic INS-1E cells, a cell line commonly used to study IAPP, decreased below 20% of control at 25 µM N21Q, whereas IAPP fibrils showed no toxicity (Fig. 1c). IAPP-mediated toxicity is associated with numerous cellular events, including oxidative stress, mitochondrial dysfunction, plasma membrane disruption and apoptosis[36]. Considering that caspase-3 activation is an upstream event to viability measured by the metabolic activity, cells were incubated for 1 to 6 h with fibrils before measuring caspase-3 activity. N21Q fibrils induced a significant increase of caspase-3 activation, with roughly a 3-fold increase compared to IAPP fibrils after 150 min (Fig. 1d). Cytotoxicity of N21Q fibrils was further evaluated using the Live/Dead assay. INS-1E β-cells treated with IAPP fibrils showed a similar green/red ratio to the cells treated with the control (Fig. 1e, Supplementary Fig. 1). The red fluorescence correlates with loss of plasma membrane integrity and the green fluorescence is associated with intracellular esterase activity. In sharp contrast, cells treated with N21Q fibrils were all red-positive, indicative of perturbation of the plasma membrane.

We investigated if the unexpected toxicity of N21Q fibrils could result from the presence of soluble oligomers associated with an incomplete fibrillization and/or dissociation of oligomers from the fibrils. First, peptides were assembled for up to 5 days and cytotoxicity of the aggregation mixtures were periodically evaluated. Prolonging self-assembly did not modify the divergent toxicity observed between WT and N21Q fibrils (Supplementary Fig. 2). Secondly, fibrils were isolated by centrifugation to remove

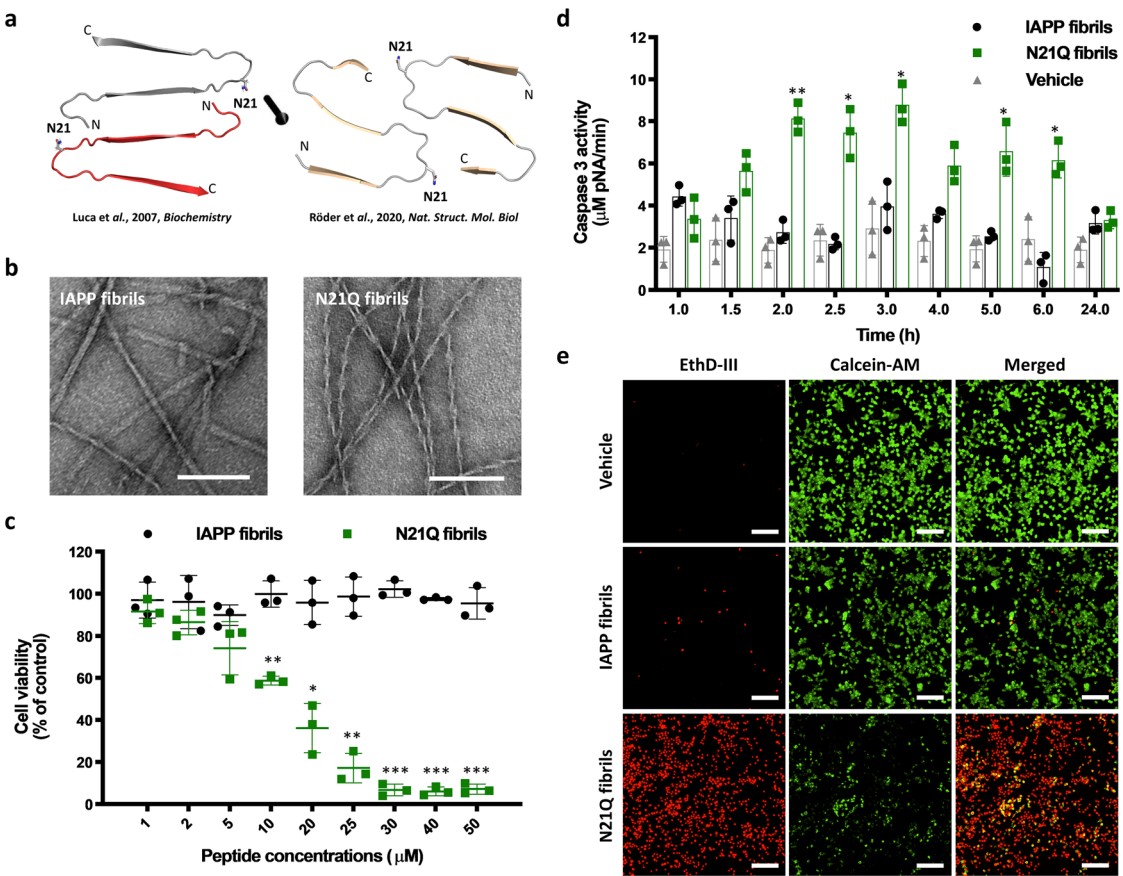

**Fig. 1 Modulating the cytotoxicity of IAPP fibrils with an N-to-Q substitution at position 21. a** SS-NMR and cryo-EM molecular models of IAPP secondary structure in amyloid fibrils showing the side chain of Asn21[29, 31]. **b** Representative TEM images of IAPP and N21Q fibrils. Scale bar: 100 nm. **c** INS-1E cells were incubated for 24 h with pre-assembled fibrils. **d** Caspase-3 activation over treatment time of INS-1E cells incubated with 50 µM fibrils. **c**, **d** Data represent mean ± S.D. of at least three independent experiments performed in triplicate. **e** Representative fluorescence microscopy images showing the distribution of live (green) and dead (red) INS-1E cells after treatment with 50 µM fibrils for 24 h. Scale bar: 50 µm. **b**–**e** Fibrils were assembled from freshly dissolved monomerized peptides incubated under quiescent conditions for 48 h at a concentration of 150 µM in 20 mM Tris-HCl buffer, pH 7.4.

any remaining soluble species that could cause cell death. Fibrils were subjected, or not, to a 30 min sonication, before being centrifuged at $35{,}000 \times g$ for 45 min. Pellets were washed, sonicated or not, and centrifuged a second time. Supernatants and pellets were analysed by transmission electron microscopy (TEM) and cytotoxicity was evaluated. N21Q fibrils isolated in the pellet, with or without sonication, remained toxic to pancreatic cells (Supplementary Fig. 3). Thirdly, as amyloid fibrils can disassemble and release oligomers[37], we evaluated the thermal unfolding of the fibrils by circular dichroism (CD) spectroscopy. According to the β-sheet signal at 218 nm, IAPP and N21Q fibrils in presence of 4 M urea showed a similar thermal unfolding midpoint ($T_m$); 58.1 ± 0.2 °C for IAPP and 61.3 ± 0.5 °C for N21Q (Supplementary Fig. 4). Fourthly, stability against proteolysis was assessed by subjecting the assemblies to 120 U/mL of proteinase K (PK) for 1 h and digestion was evaluated by MALDI-TOF mass spectrometry. N21Q and WT fibrils showed an equivalent stability against PK proteolysis, suggesting a similar degree of fibril compactness and undetectable dissociation of soluble species (Supplementary Fig. 5). These observations indicate that the cytotoxicity of N21Q assemblies is induced by defined fibrils and is not associated with presence of soluble oligomers or fibril dissociation.

**Supramolecular characterization of N21Q toxic fibrils.** To obtain insights into the molecular basis of the discrepancy in

toxicity between IAPP and N21Q fibrils, their (supra)molecular characteristics were compared. As observed by TEM and atomic force microscopy (AFM), both peptides assembled at 150 µM for 48 h formed two major distinctive fibril morphologies: flat ribbons and twisted filaments with varied pitches (Fig. 2a, b). Morphological heterogeneity of IAPP fibrils prepared in vitro is well-known[29]. Quantification of over 3000 fibrils using AFM images revealed that the prevalence of twisted fibrils in the N21Q samples was significantly higher (Fig. 2c). N21Q fibrils were somewhat shorter, with an average length of 1.08 ± 0.17 µm, compared to 1.42 ± 0.36 µm (Fig. 2c, Supplementary Fig. 6). The average height of N21Q fibrils was approximatively half of the diameter measured for IAPP fibrils, i.e. 3.12 ± 1.24 nm vs 5.98 ± 2.62 nm (3.07 vs 6.11 nm, considering the median), suggesting that the number of protofilaments composing the fibrils and/or their compactness diverge. Measurement of Young's modulus revealed that N21Q fibrils displayed lower stiffness. Structural organization was probed by attenuated total reflection Fourier transform infrared (ATR-FTIR) and far-UV CD spectroscopy. The amide I region of IAPP and N21Q spectra (1700–1600 cm$^{-1}$) were practically identical with three characteristic amide peaks (Fig. 2d). The first peak at 1685 cm$^{-1}$ suggests an anti-parallel β-sheet, in agreement with the molecular organization of IAPP fibrils[29,31]. Signal at 1645 cm$^{-1}$ represents turn structure, whereas the peak at 1616 cm$^{-1}$ is indicative of β-sheets under the amyloid fold[38]. CD spectra of IAPP and N21Q fibrils displayed a single minimum at 218 nm, corresponding to a β-sheet-rich secondary

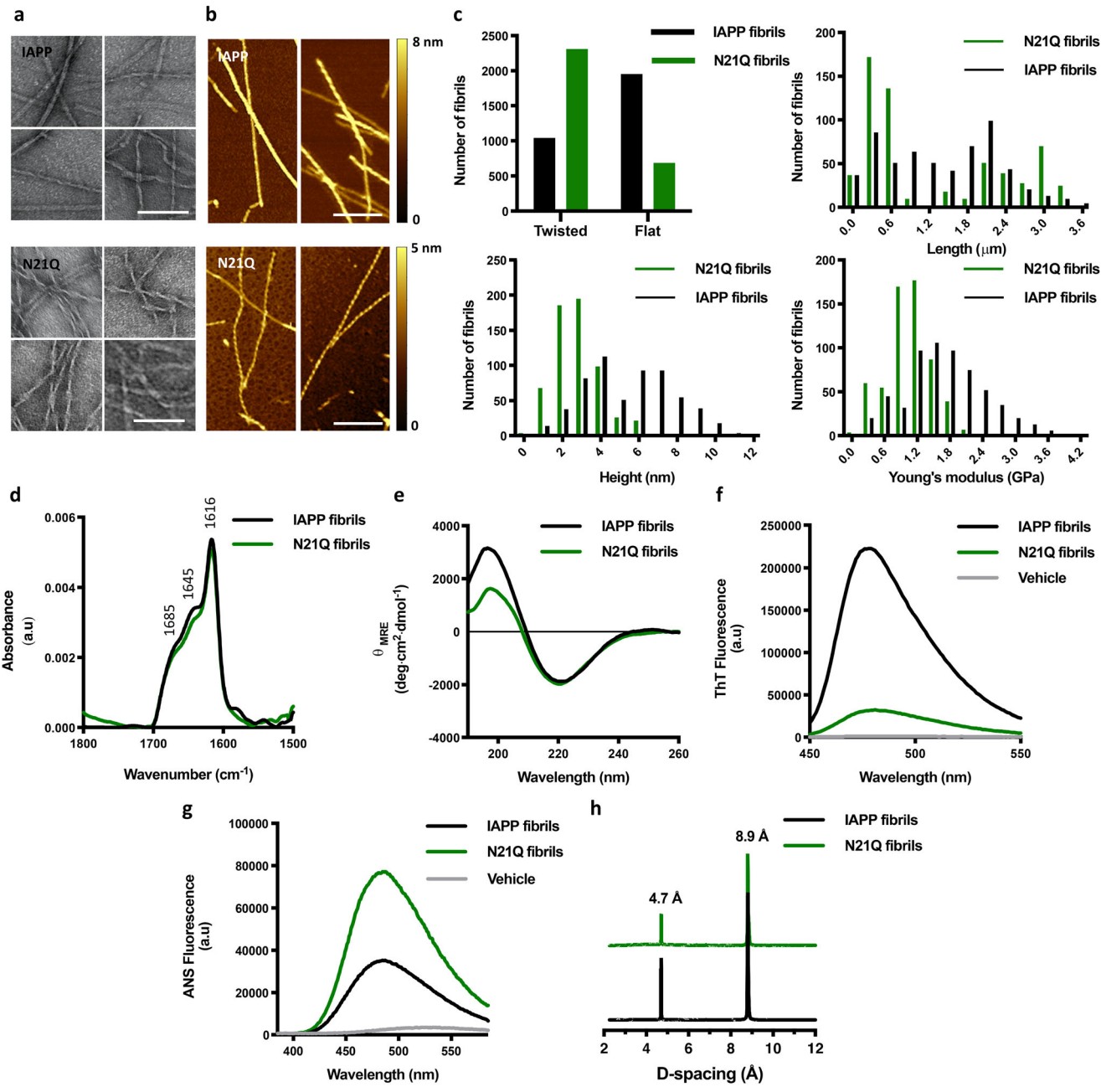

**Fig. 2 Characterization of N21Q cytotoxic fibrils. a, b** Representative images of flat and twisted IAPP and N21Q fibrils revealed by **a** TEM and **b** AFM. Scale bar: 100 nm. **c** Distributions of mesoscopic morphology and stiffness of over 3000 fibrils per peptide imaged by AFM. **d** ATR-FTIR absorbance spectra of the amide I region (1500–1800 cm$^{-1}$) of IAPP and N21Q fibrils. **e** Far-UV circular dichroism spectra of IAPP and N21Q fibrils. **f** ThT fluorescence of fibrils. Emission spectra of ThT (40 µM) with an excitation at 440 nm. **g** Surface hydrophobicity measured by ANS fluorescence. Emission spectra of ANS (100 µM) with an excitation at 355 nm. **h** Powder XRD diffraction spectra revealing a periodic amyloid packing for IAPP and N21Q fibrils. **a–h** Fibrils were assembled from freshly dissolved monomerized peptides incubated under quiescent conditions for 48 h at 150 µM in 20 mM Tris–HCl buffer, pH 7.4. Assemblies were diluted to 10 µM (**a–c**) and 50 µM (**e–g**), immediately before analysis.

structure (Fig. 2e). The amyloid architecture was probed by measuring the fluorescence of thioflavin T (ThT), whose emission increases sharply upon binding to cross-β quaternary structure[39]. Whereas fibrils assembled from IAPP presented a sharp ThT signal, N21Q fibrils led to a modest increase of fluorescence (Fig. 2f), suggestive of less defined ThT binding sites. Although low-ThT binding is generally ascribed to prefibrillar assemblies, this characteristic has been reported for fibrils assembled from different peptides, including pufferfish IAPP[40] and the Japanese mutant of Aβ (ΔE22)[41]. Accessibility of hydrophobic clusters was

evaluated using the 8-anilino-1-naphtalenesulfonic acid (ANS) that binds to solvent-exposed hydrophobic domains, leading to an increase and blue shift of fluorescence emission. In comparison to WT fibrils, the N21Q fibrils led to a strong increase of ANS fluorescence, indicative of higher surface hydrophobicity (Fig. 2g). Powder X-ray diffraction (XRD) revealed a diffraction pattern characterized by two sharp peaks and Bragg reflections corresponding to 4.7 Å and 8.9 Å periodic spacing were measured (Fig. 2h). The 4.7 Å meridional reflection corresponds to the prototypical signature of amyloids arising from the β-strand

spacing. While the 8.9 Å inter-sheet distance is rather short for amyloids, which is typically around 10 Å, the inter-sheet spacing is shorter in dry interface[42], as for powder XRD. Overall, these results indicate that IAPP and N21Q fibrils exhibit a similar structural organization, although dissimilarities were observed in the mesoscopic morphology and in the ability to interact with dyes.

SS-NMR was used to evaluate changes in the microenvironment of specific residues induced by the N-to-Q substitution that could reveal the molecular fingerprints of toxicity. Peptides, with uniformly labelled $^{13}$C and $^{15}$N residues at positions A13, F23, and V32, were assembled at 375 µM for 48 h and fibrils were recovered by centrifugation and lyophilized. Fibrils conserved their distinctive biophysical properties and cytotoxicity (Supplementary Fig. 7). Labelled residues were selected to comprise the different segments according to SS-NMR and cryo-EM models of fibrils (Fig. 3a)[29,31]. Cross-polarization (CP) dipolar assisted rotational resonance (DARR) spectra of IAPP (black) and N21Q (green) fibrils were overlaid (Fig. 3b). Assignments of residue chemical shifts were established based on the connectivity pattern and the typical ranges for each carbon. Secondary $^{13}$C chemical shifts ($\Delta\delta$) for Cα, Cβ, and CO of labelled residues were calculated as $\Delta\delta = \delta_{obs} - \delta_{RC}$ (Supplementary Fig. 8). Chemical shifts indicated that the cross-β-sheet structure is retained in both fibril preparations, as indicated by $\Delta\delta$ of Cβ being positive, and CO and Cα being negative. Spectra obtained from both fibrils were similar in the aliphatic and aromatic regions, with two minor changes regarding A13 Cβ and V32 Cγ. Pronounced changes were observed in the carbonyl region of the spectra. N21Q fibrils carbonyl resonances were deshielded compared to IAPP, indicative of either a change in structure or local environment (Fig. 3c, Supplementary Fig. 8). F23 exhibited the largest change in CO shift, from 169.8 ppm in IAPP to 173.4 ppm in N21Q fibrils, leading to alteration of F23 CO secondary shift, from strongly negative (−3.3 ppm) in IAPP to 0.3 ppm in N21Q. Considering the lack of changes in the Cα and Cβ secondary shifts of F23, this is not likely related to a major alteration in secondary structure, but may instead be a result of a local increase in hydrophobicity[43] and/or from altered hydrogen-bond interactions[44]. Change in V32 CO secondary shifts, from negative value towards zero, suggest the possibility that a similar effect was occurring around the C-terminal region.

**Kinetics of self-assembly and time-resolved cytotoxicity.** The observed distinct biological properties, i.e. cytocompatible *vs* toxic, from fibrils assembled from closely related peptides, *i.e.* IAPP *vs* N21Q, could arise from divergent aggregation pathways. Accordingly, we evaluated the kinetics of self-assembly by ThT and ANS fluorescence, as well as using an assay based on fluorescein arsenical hairpin (FlAsH). Although a weak ThT signal was measured for N21Q, typical sigmoidal traces characterized with three distinct phases (lag, elongation, saturation) were observed, suggestive of a nucleation-dependent polymerization (Fig. 4a, Supplementary Fig. 9). Lag-times of 9.9 ± 2.0 h and 7.0 ± 1.5 h were extracted respectively from IAPP and N21Q ThT kinetics, indicating that the N21Q substitution hastens nucleation. Considering the low ThT-signal of N21Q, the fluorogenic probe FlAsH was used. FlAsH, whose fluorescence quantum yield dramatically increases upon its binding to a tetracysteic tag[45], has been recently used to detect IAPP self-assembly through the formation of a non-contiguous tetra-Cys binding motif involving the N-terminal C2 and C7[46]. This method is well-suited to detect ThT-negative fibrils, as those assembled from N21Q. Self-assembly monitored by FlAsH and performed under reducing conditions, revealed a typical sigmoidal growth with lag-time of

7.3 ± 1.4 h and 2.6 ± 1.9 h for IAPP and N21Q, respectively (Fig. 4b, Supplementary Fig. 9). Kinetics of aggregation monitored by ANS fluorescence confirmed that the N21Q substitution accelerates nucleation (Fig. 4c). Gradual augmentation of the molar ratio of N21Q into IAPP self-assembly (from 1 to 10%) progressively hastened nucleation and led to reduced final ThT fluorescence and increased final ANS fluorescence, while the opposite effect was observed for the reverse experiment, i.e. IAPP into N21Q assembly reaction (Supplementary Fig. 10). These observations suggest that IAPP and N21Q monomers co-assemble, leading to fibrils that progressively acquire the characteristics of their co-assembling counterpart.

Next, the toxicity of the proteospecies assembled along the aggregation pathway was evaluated. When freshly dissolved monomerized peptides (0 h) were immediately applied to INS-1E cells, a concentration-dependant toxicity was observed for both peptides, albeit N21Q was significantly more toxic (Supplementary Fig. 11). The higher toxicity of monomeric N21Q correlated with a hastened capacity to induce caspase-3 activation and to perturb synthetic large unilamellar vesicles (LUVs) composed of phosphocholine/phosphoglycerol (DOPC/DOPG) (Supplementary Fig. 11). For time-resolved toxicity, peptides were incubated at 150 µM and after different incubation periods, the aggregation mixture was characterized by CD spectroscopy, ThT and ANS fluorescence, and TEM, and the toxicity of the proteospecies was evaluated by monitoring the viability of INS-1E cells after 5 h incubation with 50 µM pre-assembled proteospecies. As previously reported[7], WT IAPP cytotoxicity correlated with self-assembly time; the proteospecies populating the lag phase decreasing cell viability and the fibrils, i.e. over 12 h self-assembly time, being non-cytotoxic (Fig. 4d). In sharp contrast, N21Q species of the elongation and saturation phases remained cytotoxic, although their toxicity was less compared to the one induced by the assemblies of the lag phase. In agreement with the self-assembly kinetics, the random coil-to-β-sheet secondary structural conversion occurred after 2 h for N21Q and after 6 h for IAPP (Fig. 4e). ThT and ANS fluorescence reached the plateau after 6 h incubation for N21Q and well-defined filaments were observed by TEM imaging after only 2 h incubation (Fig. 4f–h), i.e. when cytotoxicity remains very high. In sharp contrast for IAPP, ThT and ANS signal reached the plateau after 12–24 h incubation and defined fibrils could be observed by TEM after 6–12 h. These data suggest that the N21Q substitution hastens IAPP self-assembly into fibrillar assemblies, and these N21Q fibrils conserve high toxicity upon elongation.

**N21Q fibrils share conformational characteristics with pre-fibrillar oligomers.** Time-resolved analysis suggested that the toxic oligomer-like characteristics of N21Q are maintained upon elongation. This hypothesis was evaluated using conformational antibodies that specifically recognize distinct supramolecular structures[47]. As revealed by dot blot analysis, isolated IAPP and N21Q fibrils were recognized by the anti-amyloid LOC and 4G8 antibodies (Fig. 5a). The oligomer-specific A11 antibody recognized oligomers, which were prepared by incubating IAPP at 150 µM for 15 min (Supplementary Fig. 12). No binding of the A11 antibody to freshly dissolved monomerized peptides was observed (Supplementary Fig. 13). Strikingly, the A11 antibody recognized N21Q fibrils (Fig. 5a). ELISA confirmed these results (Supplementary Fig. 14). In contrast to IAPP fibrils, immunogold TEM images revealed that N21Q fibrils were simultaneously recognized by the anti-amyloid 4G8 (10 nm gold-particles, white arrows) and the anti-oligomer A11 (20 nm gold-particles, yellow arrows) antibodies (Fig. 5b). IAPP and N21Q fibrils separately incubated with A11, LOC and 4GB primary antibodies

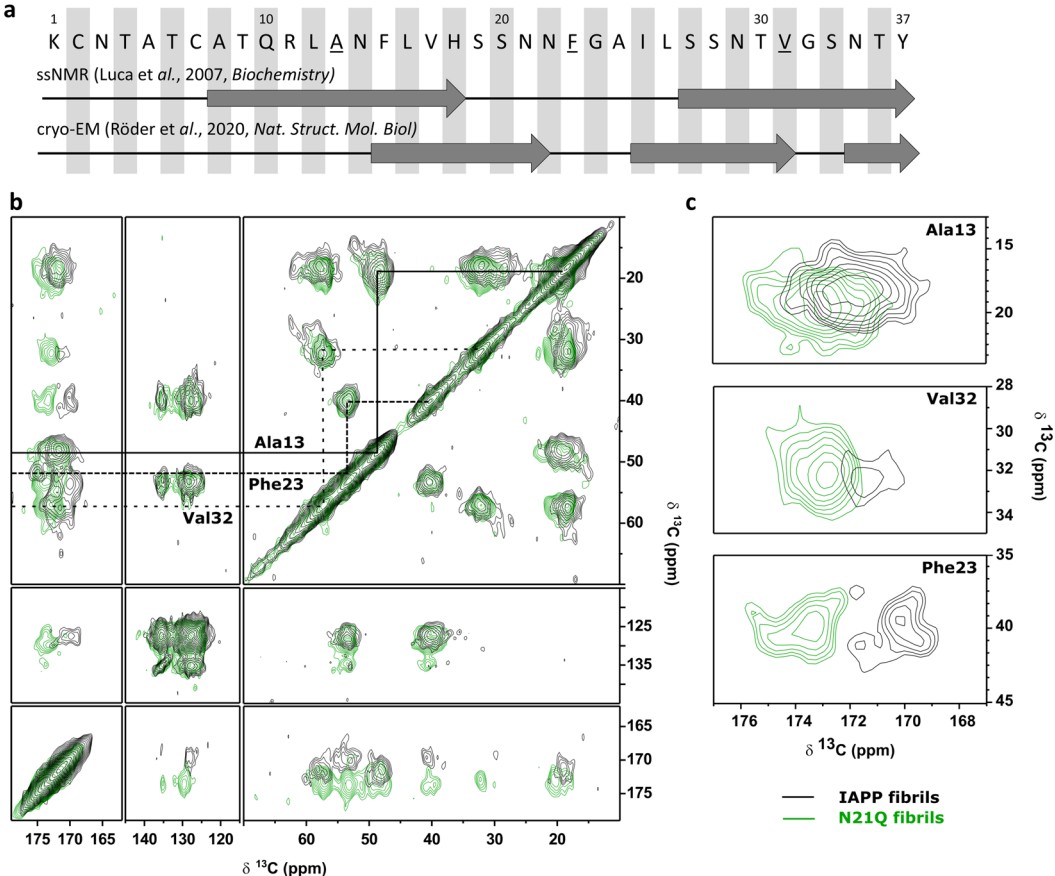

**Fig. 3 SS-NMR spectra of N21Q and WT IAPP fibrils. a** Sequence of IAPP showing the $^{13}$C and $^{15}$N uniformly labelled residues (red underlined) and the secondary structures under the amyloid fold. **b** CP-DARR spectra of IAPP and N21Q fibrils. **c** Carbonyl region of the spectrum showing changes in Ala13, Val32 and Phe23 chemical shifts. **b**, **c** Fibrils were assembled from monomerized peptides incubated under quiescent conditions for 48 h at 375 µM in 20 mM Tris-HCl buffer, pH 7.4. Fibrils were recovered by ultracentrifugation at 100,000 × $g$ for 45 min, 4 °C. Pellets were re-suspended in water and lyophilized before SS-NMR analysis.

confirmed the orthogonality of the double labelling approach (Supplementary Fig. 15). Thus, N21Q peptide preserves the A11-specific epitope upon elongation into fibrils. Several conformational models have been proposed for IAPP oligomers, including α-helix coiled-coil[48], β-hairpin assemblies[49] and parallel β-sheets involving the 20–29 segment[50]. The latter model proposes that the 20–29 segment of IAPP mediates initial self-recognition and that the oligomers contain stacks of solvent-exposed β-sheets involving the $^{23}$FGAIL$^{27}$ region. Considering that N21Q fibrils exhibit oligomeric characteristics, exposure of the 20–29 region during self-assembly was evaluated by replacing F23 by a p-cyano-phenylalanine (F$_{CN}$) whose fluorescence is modulated by solvent exposure. As previously reported[51], fluorescence of F23F$_{CN}$ IAPP decreased over self-assembly time, indicating that position 23 is being buried with fibril growth (Fig. 5c). In sharp contrast, high fluorescence persisted during F23F$_{CN}$ N21Q self-assembly. This observation suggests that the $^{23}$FGAIL$^{27}$ region remains solvent-exposed in the fibrils, in agreement with the SS-NMR data showing alterations of F23 chemical shifts in N21Q fibrils (Fig. 3). Solvent-exposure of residue F15 was evaluated by incorporating a F$_{CN}$ at position 15. Fluorescence of both F15$_{CN}$ peptides decreased over incubation time (Fig. 5c), suggesting that this region is buried in N21Q and IAPP fibrils. These data indicate that the oligomer conformation of N21Q is preserved upon fibril elongation and that the exposure of the FGAIL hydrophobic region likely contributes to the toxicity of N21Q fibrils.

**N21Q fibrils disrupt plasma membrane**. Several mechanisms mediating the toxicity of oligomers have been proposed, and plasma membrane disruption remains one of the most abundantly described[52]. The capacity of N21Q fibrils to disrupt lipid membranes was compared to the one of IAPP oligomers. Disruption of lipid bilayer was first evaluated using DOPC:DOPG (7:3) LUVs loaded with self-quenched calcein. WT oligomers (Supplementary Fig. 12) and N21Q fibrils induced high permeabilization of LUVs with leakage of 40–60% after 3 h incubation, whereas IAPP fibrils induced below 10% of leakage after 5 h incubation (Fig. 6a). Plasma membrane disruption of pancreatic β-cells was evaluated by measuring the release of the cytoplasmic enzyme lactate dehydrogenase (LDH). IAPP fibrils did not led to a significant release of LDH, whereas N21Q fibrils and WT oligomers induced high LDH activity in the media (Fig. 6b). IAPP oligomer mixture and N21Q fibrils induced the equivalent time-dependant activation of caspase-3, suggestive of a similar mechanism of toxicity (Fig. 6c). Interaction of N21Q fibrils with the plasma membrane was evaluated by confocal microscopy and compared with toxic oligomers and cytocompatible fibrils. Fluorescent quaternary species were prepared by co-assembling Alexa Fluor 488-labelled monomers with unlabelled monomerized peptides at a molar ratio of 0.5:99.5, allowing the formation of fluorescent fibrils and oligomers with equivalent (supra)structural and biological properties to unlabelled peptides. CHO cells were used, since INS-1E cells detached promptly in presence of N21Q fibrils, precluding the washing steps necessary

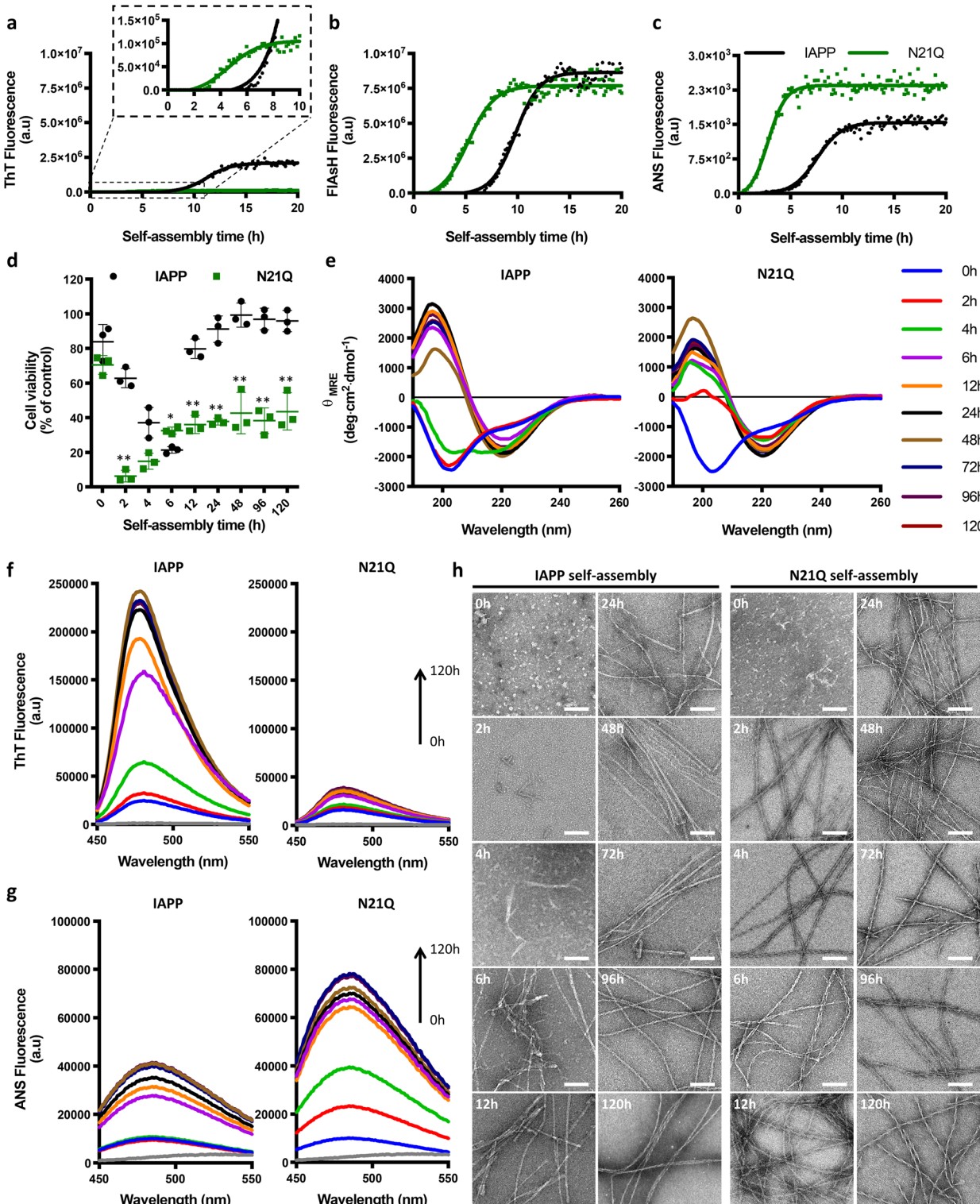

**Fig. 4 Kinetics of self-assembly and time-resolved analysis of cytotoxicity. a–c** Kinetics of self-assembly monitored by **a** ThT, **b** FlAsH, and **c** ANS fluorescence. Monomerized peptides were incubated at 12.5 μM under quiescent conditions in 20 mM Tris-HCl buffer, pH 7.4 in the presence of ThT (40 μM), FlAsH (0.5 μM) or ANS (50 μM). Fluorescence of ThT (Ex 440 nm, Em 485 nm), FlAsH (Ex 508 nm, Em 533 nm) and ANS (Ex 355 nm, Em 480 nm) was measured every 10 min. Data from triplicates were averaged and fitted with a Boltzmann sigmoidal curve. **d** Time-resolved cytotoxicity of proteospecies evaluated by measuring the metabolic activity of INS-1E upon 5 h incubation with 50 μM pre-assembled peptides. **e–h** Time-resolved self-assembly of IAPP and N21Q monitored by **e** CD spectroscopy, **f** ThT fluorescence, **g** ANS fluorescence and **h** TEM. **d–h** Freshly dissolved monomerized peptides were incubated under quiescent conditions at 150 μM in 20 mM Tris-HCl buffer, pH 7.4 and after the indicated time of self-assembly, the aggregation mixture was characterized and evaluated for cell toxicity.

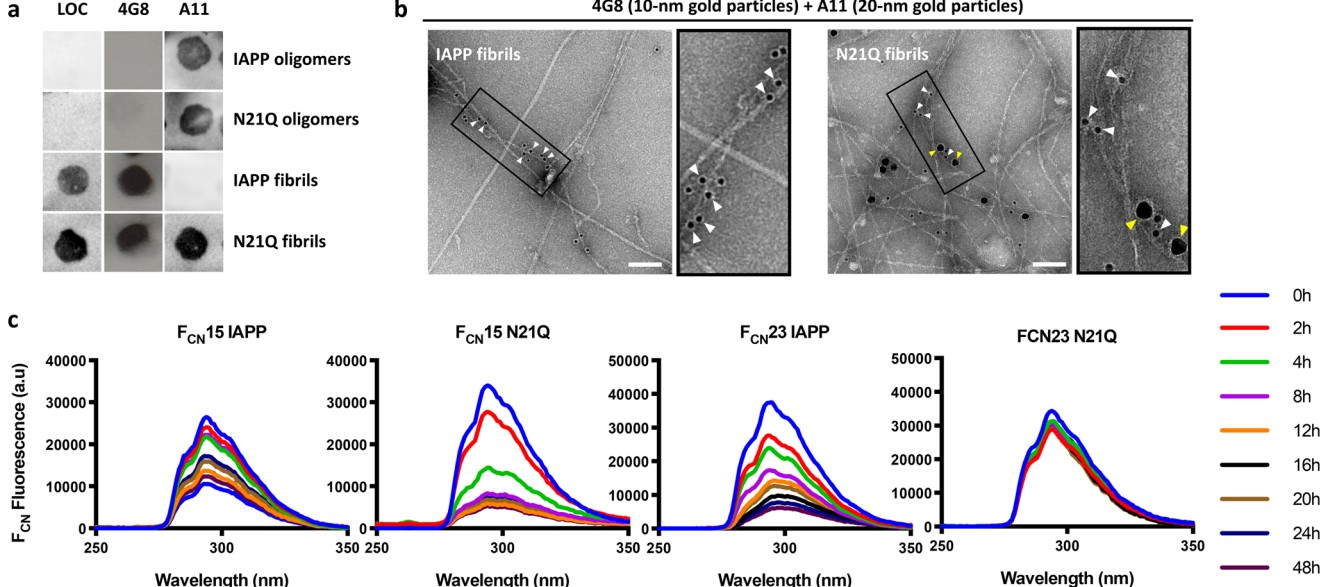

**Fig. 5 N21Q fibrils exhibit oligomer-like conformation and expose residue F23. a** Dot-blot analysis of IAPP and N21Q oligomers and fibrils. **b** Immunogold labelling electron microscopy images of IAPP and N21Q fibrils, scale bar: 100 nm. 3× zoom image of the region indicated by the rectangle. **a, b** Peptides were incubated under quiescent conditions for 15 min (oligomers) and 48 h (fibrils) at 150 μM in 20 mM Tris-HCl buffer, pH 7.4. **c** $F_{CN}$ fluorescence emission spectra over self-assembly time. Peptides were incubated under quiescent conditions for 48 h at 150 μM in 20 mM Tris-HCl buffer, pH 7.4.

for microscopy analyses. CHO cells showed similar susceptibility towards concentration-dependant membrane disruption and toxicity induced by the proteospecies (Supplementary Fig. 16). Z-stack reconstruction images revealed that IAPP fibrils gather at the cell surface and do not incorporate within the plasma membrane (Fig. 6d). In sharp contrast, IAPP oligomers colocalized with lipids and extracted them from the membrane. Strikingly, N21Q fibrils damaged plasma membrane by pulling-out lipids, as observed with the formation of large yellow puncta containing the fluorescent fibrils in green and lipid vesicles in red (Fig. 6d). These data reveal that the N21Q fibrils mediate plasma membrane disassembly and permabilization, perhaps an upstream event to cell death, as observed for toxic oligomers.

**The oligomer-like conformation of N21Q fibrils propagates to IAPP.** It is known that the conformation of an aggregate can propagate to a naive polypeptide building block, whose assemblies will ultimately acquire the properties of the pre-formed aggregates. This could occurred through secondary nucleation, which corresponds to the catalyzed nucleation of monomers or oligomers by pre-assembled fibrils[53], and/or through the elongation of pre-existing fibrillar seeds by the addition of monomers/oligomers to the growing end. We evaluated if the properties of the N21Q fibrils can be propagated to WT IAPP by supplementing IAPP self-assembly reaction with pre-assembled N21Q fibrils. Interestingly, increasing the molar ratio of N21Q seeds, from 0.5 to 5% (molar equivalent), in IAPP self-assembly induced a progressive decrease of ThT fluorescence, an increase in surface hydrophobicity, and the formation of a conformational epitope that is recognized by the A11 antibody (Fig. 7a, b). Appropriate controls validated that 2.5 μM N21Q seeds (5 mol%) do not contribute to A11 binding and ANS signal (Supplementary Fig. 17). Next, IAPP labelled at position F23 ($^{13}$C, $^{15}$N) was incubated with unlabelled N21Q fibrils and the CO chemical shift of the seeded fibrils was compared with homogenous labelled F23 ($^{13}$C, $^{15}$N) IAPP and N21Q fibrils. The carbonyl region of the CP-DARR spectra revealed that the CO chemical shift of F23 was progressively deshielded as the ratio of N21Q seeds increased

(Fig. 7c), indicative of a change in the environment of F23 towards N21Q conformation. Moreover, incubating monomeric F23$F_{CN}$ IAPP in presence of N21Q seeds led to the formation of F23$F_{CN}$ IAPP fibrils with high solvent exposure of F23 side chain, showing that the N21Q conformation propagates to WT peptide (Fig. 7d). TEM images revealed that the presence of N21Q fibrils within IAPP self-assembly mixture led to the formation of a mixture of twisted fibrils and flat ribbons, with twisted morphology being progressively more prevalent with increasing N21Q seeds (Supplementary Fig. 18). Predominantly, IAPP fibrils grown in presence of over 2% of N21Q seeds were toxic to pancreatic β-cells (Fig. 7e). As control, INS-1E cells were treated with the respective concentrations of N21Q seeds (0.25–2.5 μM), and no decrease of cell viability was measured under these concentrations (Supplementary Fig. 19). The inverse experiment was also performed, i.e. N21Q self-assembly was seeded with IAPP fibrils. Under these conditions, N21Q assemblies progressively acquired the properties of IAPP fibrils; low toxicity, high ThT-binding, low hydrophobicity, progressive burying of Phe23, low recognition by the A11 antibody, and higher prevalence of ribbon morphology (Supplementary Fig. 20). These results indicate that the conformation of N21Q fibrils that is recognized by the A11 antibody is transmissible, which implies that the cytotoxic motif constitutes an organized supramolecular architecture.

## Discussion

Inconsistency among the reported toxicity of the proteospecies of the amyloid cascade has been associated with their high polymorphism and/or dynamic transient nature[54,55]. Site-specific mutagenesis can lead to the formation of assemblies with discrete morphologies and each of these structural ensembles can exert distinct biological functionalities. Herein, we took advantage of a trivial N-to-Q substitution to identify a fibrillar nanostructure with oligomer-like characteristics and high cytotoxicity, underlining the high susceptibility of amyloid self-assembly to point mutation. Toxicity of defined and mature amyloid assemblies has been ascribed to fibril breakage and leakage of soluble oligomers. This is likely not the case for N21Q fibrils, as these assemblies

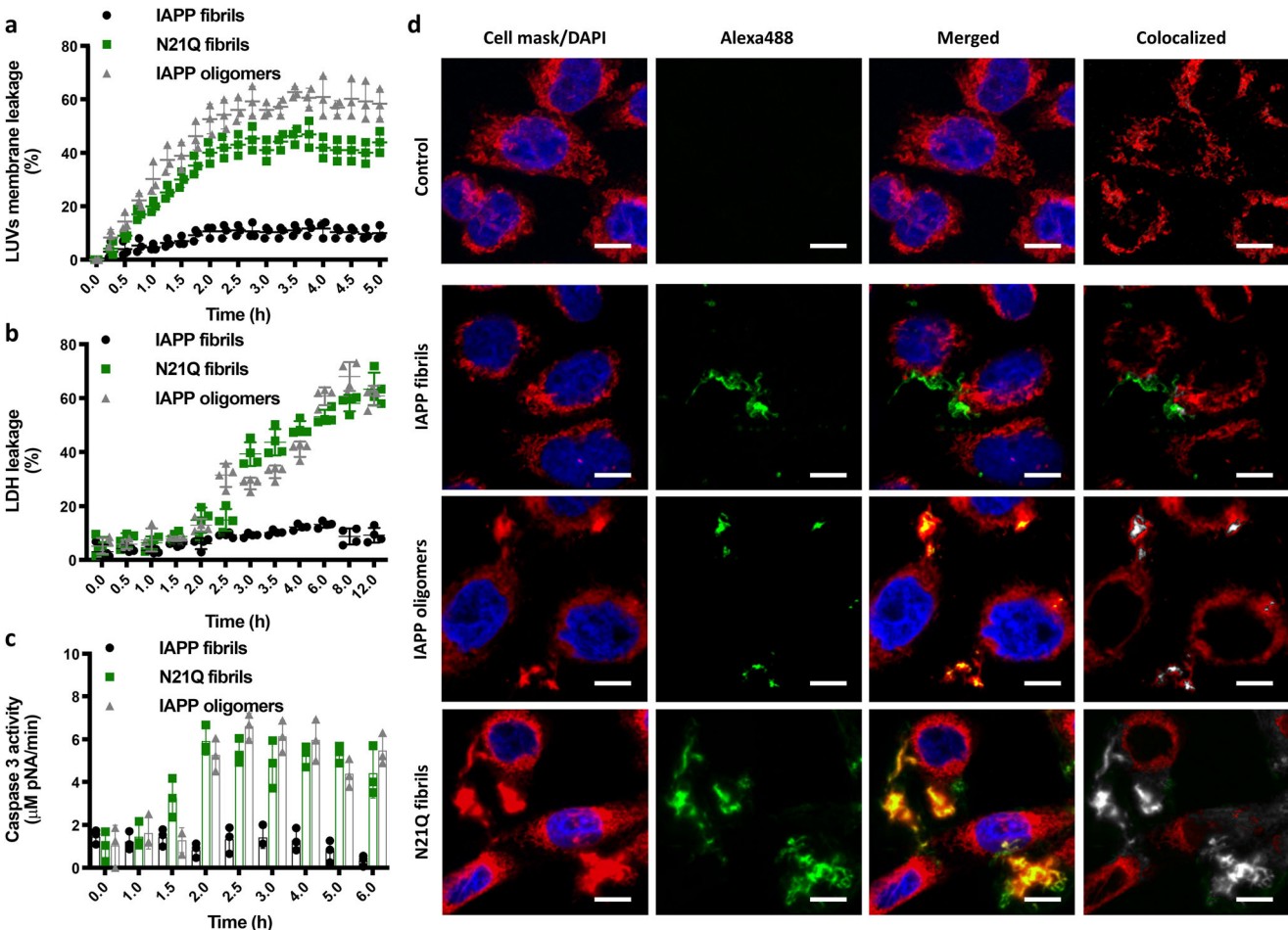

**Fig. 6 N21Q fibrils disrupt lipid vesicles and plasma membrane. a** Membrane leakage of 500 µM DOPC:DOPG LUVs (7:3) by 50 µM oligomers and fibrils. **b** LDH release from INS-1E cells incubated with 50 µM fibrils and oligomers for different times. **c** Caspase-3 activation upon incubation of INS-1E cells with 50 µM fibrils and oligomers for different treatment times. **d** Representative fluorescent confocal images of CHO-K1 cells treated with Alexa Fluor-488 labelled (green) fibrils and oligomers. Cell nuclei and plasma membrane were stained in blue and red, respectively. Scale bar: 10 µm. **a**, **b** Oligomers and fibrils were assembled from monomerized peptides incubated under quiescent conditions for 15 min (oligomers) and 48 h (fibrils) at 150 µM in 20 mM Tris-HCl buffer, pH 7.4.

showed comparable thermal and enzymatic stability with cytocompatible IAPP fibrils. The oligomer properties of N21Q assemblies are preserved after fibril elongation and are transmissible. On the one hand, N21Q fibrils share prototypical amyloid characteristics with IAPP fibrils, including the 4.7 Å distance between stacked β-strands, a conformation rich in β-sheet, low solvent-exposition of F15, recognition by anti-amyloid antibodies, and long and linear morphology. One the other hand, cytotoxic N21Q fibrils diverge from IAPP fibrils by low ThT-binding, high surface hydrophobicity, solvent exposure of F23, recognition by A11 antibody, short thickness and higher proportion of twisted filaments. These divergent properties, which arise from the incorporation of a methylene group within N21 side chain, are associated with high ability to disrupt lipid membranes and to cause cellular dysfunction.

Previous studies have proposed cytotoxic determinants of prefibrillar species, including size, hydrophobicity, compactness, rigidity and morphology. For the Aβ peptide, it has been shown that toxicity correlates inversely to the size of aggregates[56–60]. It has been proposed that low molecular weight oligomers, with a high diffusion coefficient, diffuse rapidly within biological environment, favoring aberrant interactions with cellular components. Herein, we showed that cytotoxicity is mediated by defined fibrils with microscale length, indicating that toxicity does

not necessarily correlate with oligomerization state, suggestive of conformational-dependent toxicity. Oligomers assembled from Sup35[61], HypF-N[62], Aβ[60,63], and α-synuclein[8] with similar size and morphology can have divergent toxicity, which correlates with exposure of hydrophobic clusters and capacity to disrupt plasma membranes[64]. Poorly toxic oligomers have compact structures with the hydrophobic region buried inside, as for mature amyloid fibrils. Conversely, loose oligomers with high degree of hydrophobic exposure tend to be more cytotoxic[62,65]. In the present study, N21Q fibrils exhibited high hydrophobicity and these hydrophobic clusters likely involve organized stacks of the [23]FGAIL[27] segment. Surface hydrophobicity of N21Q assemblies promoted its insertion within plasma membrane, resulting in disassembly of lipid bilayers and membrane permeabilization. Although both fibril preparations showed polymorphism, the prevalence of twisted fibrils was significantly more important for N21Q. Moreover, as observed by AFM imaging, the height of N21Q filaments was approximatively half of the height of IAPP fibrils. Thus, the N21Q substitution likely blocks the self-assembly process into twisted protofibrils by precluding their packaging in fibrils, as previously observed for the amyloid self-assembly of the immunoglobulin light chain[66]. Aβ twisted fibrils have been shown to be more toxic to their striated ribbon counterpart[67], and twisted morphology has been observed for fibrils extracted from

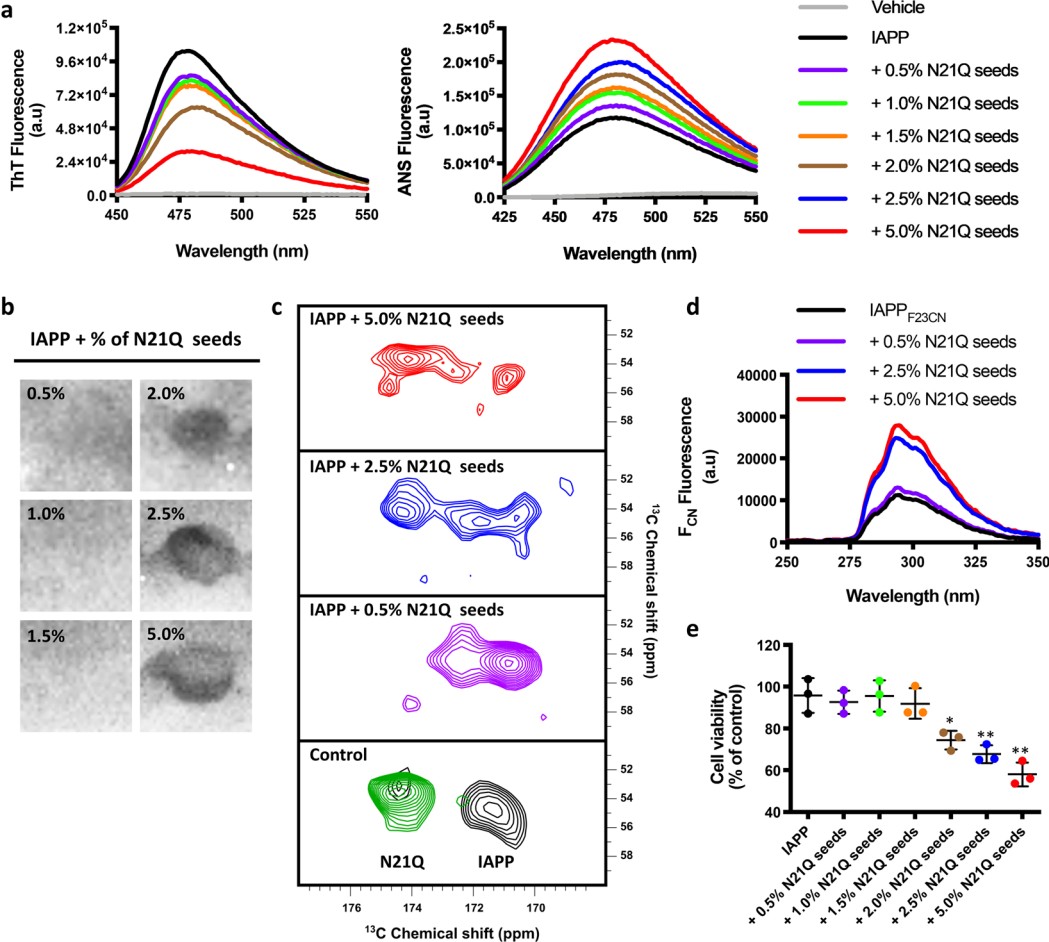

**Fig. 7 The oligomer-like conformation of N21Q fibrils propagates to IAPP. a–d** Self-assembly of IAPP seeded with N21Q pre-assembled fibrils (0.5–5 mol %) was characterized by **a** ThT fluorescence and ANS fluorescence, **b** dot-blot analysis with the anti-oligomer A11 antibody, **c** SS-NMR (CP-DARR) of F23 labelled IAPP, and **d** $F_{CN}$ fluorescence of the $F23F_{CN}$ IAPP. **e** Cell viability of INS-1E treated with 50 μM IAPP fibrils seeded with N21Q pre-assembled fibrils. **a–e** N21Q fibrillar seeds were grown for 48 h (quiescent conditions, 150 μM in 20 mM Tris-HCl buffer, pH 7.4) before being isolated and used to seed the amyloid formation of IAPP under quiescent conditions for 48 h.

Alzheimer patients[68]. The capacity of N21Q fibrils to perturb lipid membranes could be associated with twisted supramolecular organization, which increases surface hydrophobicity. Twisted morphology, either from twisting of protofilament(s) around a central axis or wrapping of protofilaments around an internal axis, emerges early during elongation. While stacking of monomers, or oligomers, onto growing fibrils is driven by hydrophobic contacts, protofilament twisting and wrapping are modulated by a fine balance of specific interactions, including steric repulsions and hydrogen bonding, both being dependent of residue side chains[69]. The N-to-Q substitution affects twisting and/or wrapping of protofilaments as well as the height and stiffness of the resulting fibrils, which could be related to changes in the strength and/or the positioning of hydrogen bonds involving residue-21.

IAPP self-assembly can be ascribed to a nucleated conformational conversion, in which hydrophobic collapse of monomers prompts oligomerization into ensemble of aggregates followed by the slow conversion of competent oligomers into protofilaments[46]. The 20–29 segment has been proposed to drive self-recognition into on-pathway oligomers, which could be composed of stacks of β-sheets[50,70]. Accordingly, nucleation is kinetically controlled by an energy barrier involving the conversion of this segment from a β-sheet into a loop, which allows formation of the hairpin[29] or S-shaped[31,32] structure within monomers of the protofilament. Not

only N21 acts as a molecular hinge modulating primary nucleation[27], it forms extensive interlayer hydrogen-bonding interactions within protofilaments, known as a polar ladder[31,32]. As observed by time-resolved ThT, ANS and FlAsH fluorescence, CD spectroscopy, and solvent exposition of $F_{CN}$15, N21Q substitution reduced the lag phase, indicating that the amide group at position 21 is a key player in nucleation and, perhaps, elongation. Particularly, the conformation recognized by the A11 antibody is preserved during elongation. As observed by the solvent-exposure of $F_{CN}$23 and SS-NMR analysis, this conformational epitope is likely composed of repeating arrays of $^{23}$FGAIL[27] that remain exposed in N21Q mature fibrils. Seeding experiments revealed that the oligomer-like conformation of N21Q fibrils is structurally defined at the quaternary level. These observations suggest that the N21Q substitution unlocks an alternative pathway emerging concurrently to primary nucleation, leading to cytotoxic fibrils.

In contrast to soluble oligomers, N21Q fibrils seeded self-assembly and the resulting IAPP assemblies were cytotoxic. This ability to seed amyloid formation indicates that the oligomer-like conformation of N21Q is organized into a repeating display of monomers, wherein all monomers share identical conformation. The transmissible and exposed lattice of monomers recognized by the A11 antibody and involving the 20–29 domain likely represents the toxic quaternary epitope. Considering that N21Q fibrils

share conformational and biological characteristics with toxic soluble oligomers, including the ability to pull out lipids from plasma membrane and disassemble lipid bilayers, these stable fibrils offer a unique opportunity to define the structural determinants of oligomer-induced cell death. The present study showed that the conformational A11 epitope can be preserved upon fibril elongation and can propagate, demonstrating that cytotoxicity of amyloids is associated with specific structural features and not uniquely to the presence of soluble oligomers. The identification of these defined filaments expands the diversity of amyloid assemblies and increases the complexity of quaternary structures associated with amyloid-related disorders.

## Methods

**Peptide synthesis, purification and sample preparation**. Peptides were synthesized on solid support using Fmoc-chemistry and 2-(6-chloro-1-H-benzo-triazole-1-yl)-1,1,3,3-tetramethylaminium hexafluorophosphate (HCTU) coupling strategy with the incorporation of oxazolidine pseudoproline derivatives[71]. After cleavage, crude peptides were purified by RP-HPLC and collected fractions were analysed by LC/MS-TOF. Disulfide bond formation between Cys-2 and Cys-7 was achieved by dimethyl sulfoxide oxidation under mild agitation. Cyclized peptides were purified by RP-HPLC and fractions corresponding to the desired product with purity higher than 95% were pooled and lyophilized. Peptides labelled with Alexa488 NHS ester were prepared as previously described[6]. Aliquots of mono-merized peptides were prepared by dissolving the lyophilized peptide in hexafluoro-2-propanol (HFIP). The solution was sonicated for 30 min, filtered through a 0.22 μm hydrophilic filter and lyophilized. The resulting peptide powder was weighted using an ultra-microbalance, solubilized for a second time in HFIP to 1 mg/ml, sonicated for 30 min and lyophilized again. Peptide concentrations were validated by measuring absorbance at 280 nm using a theoretical molar extinction coefficient of 1490 $M^{-1}$ $cm^{-1}$ and by measuring the area under curve at 229 nm by RP-HPLC. Monomerized dried samples were kept dried at −80 °C until used, but not for longer than 4 weeks.

**Preparation of oligomers and fibrils**. Prefibrillar species and fibrils were prepared by incubating freshly dissolved monomerized peptides under quiescent conditions at 150 μM in 20 mM Tris-HCl buffer, pH 7.4 for 15 min and 48 h, respectively, unless otherwise stipulated[46]. Fluorescently labelled peptide oligomers and fibrils were prepared by incubating non-labelled peptides with Alexa Fluor 488-labelled peptides (99.5:0.5, molar ratio) under quiescent conditions for 48 h at a final concentration of 150 μM.

**Fluorescence spectroscopy**. Peptide samples were diluted to a concentration of 50 μM, in presence of 40 μM of ThT or 100 μM ANS. Fluorescence was measured in a quartz ultra-micro cells 10 mm length. For ThT, excitation was set at 440 nm and emission from 450 nm to 550 nm was recorded. For ANS, the excitation was set at 355 nm and emission was recorded from 385 nm to 585 nm. Data are the average of at least four individual experiments.

**Kinetics of amyloid formation**. Solutions were prepared by dissolving the monomerized peptide in 20 mM Tris-HCl, pH 7.4. Assays were performed at 25 °C without agitation in sealed black-wall clear-bottom 96-well non-binding surface plates with a total volume of 100 μL per well. Final peptide concentrations were 12.5 or 25 μM in presence of ThT (40 μM), ANS (50 μM) or FlAsH (0.5 μM). Fluorescence of ThT (Ex. 440 nm, Em. 485 nm), ANS (Ex. 355 nm, Em. 480 nm) or FlAsH (Ex. 508 nm, Em. 533 nm) were measured from the bottom of the well every 10 min. over the course of 20 h. Data obtained from triplicate wells were averaged, corrected by subtracting the corresponding control reaction and plotted as fluorescence vs time. Data of time-dependence of fluorescence were fitted to a sigmoidal Boltzmann model where $T_{50}$ is the time required to reach half of the fluorescence intensity, $k$ is the apparent first-order constant and $Y_{max}$ and $Y_0$ are, respectively, the maximum and initial fluorescence values (Eq. 1):

$$Y = \frac{Y_0 + (Y_{max} - Y_0)}{1 + e^{-(T-T_{50})/k}} \quad (1)$$

The lag time is described as $T_{50} - 2/k$. Data (lag time and final fluorescence) of at least four different lots of peptides were averaged and were expressed as the mean ± S.D.

**Circular dichroism spectroscopy**. Peptide samples were diluted to a final concentration of 50 μM and incorporated into a 1 mm path length quartz cell. Far-UV CD spectra were recorded from 190 to 260 nm using a Jasco J-815 CD spectropolarimeter at 25 °C. The wavelength step was set at 0.5 nm with an average time of 10 s per scan at each wavelength step. Spectrum was background subtracted with peptide-free buffer. Raw data were converted to mean residue ellipticity (MRE). Thermal unfolding transitions were monitored by the variation of CD signal at 222

nm between 25 °C and 104 °C with a heating rate of 1.0 °C min$^{-1}$. Transitions were evaluated using a nonlinear least square fit assuming a two-state model[72], i.e. assembled and unassembled.

**Attenuated total reflectance-Fourier transform infrared spectroscopy**. ATR-FTIR spectra were recorded using a Nicolet Magna 560 spectrometer equipped with a nitrogen-cooled MCT detector. Each spectrum was an average of 128 scans recorded at a resolution of 2 cm$^{-1}$ using a Happ–Genzel apodization. Data analysis were performed using Grams/AI 8.0 software.

**Powder X-ray diffraction**. Fibrils were deposited on an X-ray diffraction lamella and dried overnight. Powder XRD was performed using a Bruker D8 Advance X-ray diffractometer. The current and the voltage were 40 mA and 40 mV, respectively, with a step size of 0.112° s$^{-1}$ in the 2θ range of 5–50°. Interplanar distances were determined from powder raw pattern (2θ), satisfying Bragg's condition[73].

**Transmission electron microscopy**. Fibrils were diluted to 10 μM in deionized water and immediately applied to glow-discharged carbon films on 300 mesh copper grids. After adsorption, samples were negatively stained with 1.5% uranyl formate. Images were recorded using a FEI Tecnai 12 BioTwin microscope operating at 120 kV and equipped with an AMT XR80C CCD camera system. For immunogold labelling, grids with adsorbed fibrils were floated on droplets of 4% of Tween in PBS to block non-specific binding for 30 min and washed with droplets of PBS 0.05% Tween. After incubation with primary antibodies (A11, LOC and/or 4G8) in PBS 0.05% Tween for 90 min, grids were washed before being incubated with droplets of gold-conjugated secondary antibody (goat anti-mouse IgG for 4G8, and goat anti-rabbit IgG for LOC and A11) for 1 h. Grids were washed and stained with 1.5% (w/v) uranyl formate.

**Atomic force microscopy**. Prefibrillar species and fibrils were diluted to 10 μM in 1% acetic acid, and immediately applied to freshly cleaved mica. The mica was washed twice with deionized water and air-dried. Images were acquired on a Veeco/Bruker Multimode AFM using tapping mode with a silicon tip (2–12 nm tip radius, 0.4 N/m force constant) on a nitride lever. Images were taken at 0.2 Hz and 1024 scan/minute.

**Solid-state nuclear magnetic resonance**. Fibrils for SS-NMR were prepared using uniformly labelled $^{13}$C, $^{15}$N amino acids at positions Ala13, Phe23 and Val22. Fibrils assembled at 375 μM for 48 h were concentrated by ultracentrifugation at 100,000 × $g$ for 45 min at 4 °C. The supernatant was discarded and the pellet was resuspended in water and lyophilized. The lyophilized fibrils were then packed dry into NMR rotors, with a final recovery ranging between 50 and 80% of the mass of the initial peptide. For IAPP fibrils, approximately 50% of the starting mass was recovered, whereas for N21Q fibrils, almost 80% of the starting mass was recovered. NMR spectra were recorded on a 400 MHz wide-bore Bruker Avance III-HD using a triple-resonance 1.9 mm magic-angle spinning (MAS) probe operating in double resonance. MAS speed was set at 26 kHz for 2D DARR experiments with a 200 ms mixing time and the $^{13}$C and $^1$H radiofrequency fields were 83 kHz and 100 kHz, respectively. Cross-polarization was used (1.3 ms) during acquisition with a 70–100% intensity ramp on the $^1$H radiofrequency channel, and protons were decoupled using the spinal-64 sequence. All experiments were recorded at 277 K and spectra were externally referenced using adamantane, setting the $^{13}$C methylene (CH$_2$) signal to 38.48 ppm, relative to tetramethylsilane (TMS)[74]. Data were acquired and processed using Topspin and Mestrenova.

**Dot blot**. Ten μl of peptide sample (150 μM) was applied to a nitrocellulose membrane. The membranes were allowed to air dry and were blocked in 5% non-fat dried milk in TBS-T for 30 min at RT. Membranes were incubated with the primary antibodies (A11, LOC, 4G8) in TBS-T. After three washes in TBS-T, membranes were incubated with HRP-conjugated secondary antibody (goat anti-mouse IgG for 4G8, goat anti-rabbit IgG for LOC and A11 antibodies) for 1 h. The secondary antibodies were diluted 1:10,000 in 5% non-fat dried milk in TBS-T. Blots were washed three times with TBS-T, and secondary antibody reactivity was revealed by enhanced chemioluminescence.

**Cell viability**. INS-1E cells (Millipore Sigma) were plated in black wall clear bottom 96-well plates at a density of 20,000 cells/well (100 μl/well) in RPMI-1640 medium supplemented with 10% FBS, 2 mM L-glutamine, 100 U/ml penicillin, 100 mg/ml streptomycin, 10 mM HEPES, 1 mM sodium pyruvate, 50 mM β-mercaptoethanol. After 48 h incubation at 37 °C in a 5% CO$_2$, cells were treated by the addition of 50 μl of peptides (monomers, prefibrillar or fibrils) at 3× of final concentrations (50 μM to 1 μM) solubilized in 20 mM Tris, pH 7.4. Cells were incubated for 5 h (for time-resolved assay) or 24 h and cellular viability was measured by the resazurin reduction assay. For CHO-K1 (ATCC), cells were plated at a density of 20,000 cells/well (100 μl/well) in Ham F-12 medium supplemented with 10% FBS, 2 mM L-glutamine, 100 U/ml penicillin, 100 mg/ml streptomycin. Cell viability (in %) was calculated from the ratio of the fluorescence of the treated sample to the control cells (non-treated). Data (in %) of at least four lots of

peptides were averaged and were expressed as the mean ± S.D. For Live/Dead assays, cells were plated in 12-well plates at density of 150,000 cells/well for 48 h before treatment. Analyses were performed by the addition of the reagent solution (4 μM of ethidium homodimer-1; 2 μM of calcein-AM) and after 45 min incubation, cells were imaged by fluorescent microscopy.

**Caspase-3 activity assay**. Cells were cultured in 12-well plates at a density of 400,000 cells/well. After 48 h incubation, cells were treated with peptide solutions for different time (0–24 h), at final peptide concentration of 50 μM. Cells were lysed on ice for 30 min followed by a centrifugation at $16,000 \times g$ for 20 min. Caspase-3 activity in the supernatant was measured by a colorimetric assay and the caspase-3 activity was expressed in μmol pNA released per min per ml of cell lysate. Data of at least three individual experiments performed in triplicate were averaged and were expressed as the mean ± S.D.

**Lactate dehydrogenase (LDH) release assay**. INS-1E and CHO-K1 cells were seeded in 96-well plates at 30,000 cells/well. After 48 h incubation, cells were treated with peptide solutions. Media (50 μl) was collected at different times (0–12 h) and incubated with the reaction mixture from the Pierce LDH Cytotoxicity Detection Kit for 30 min at room temperature. The reaction was stopped and absorbance was measured from 490 to 680 nm. To determine LDH activity, the 680 nm absorbance value (background) was subtracted from the 490 nm absorbance before calculation. Percentage of LDH leakage was calculated (Eq. 2):

$$\%LDH\ leakage = \frac{LDH\ activity_{peptide-treated} - LDH\ activity_0}{LDH\ activity_{max} - LDH\ activity_0} \times 100 \qquad (2)$$

where $LDH\ activity_0$ is the absorbance measured in absence of peptide and $LDH\ activity_{max}$ is the absorbance measured for cells treated with lysis buffer. Data of at least three individual experiments performed with different lot of peptides were averaged and expressed as the mean ± S.E.M.

**Leakage of large unilamellar vesicles**. DOPC/DOPG (7:3, molar ratio) lipids were solubilized in chloroform in a glass tube and solvent was evaporated with a nitrogen stream. Lipid film was rehydrated in a 20 mM Tris-HCl pH 7.4 buffer containing 70 mM calcein for 30 min. Solution was freeze-thawed five times before being extruded with a 100 nm polycarbonate membrane for 20 cycles. Free calcein was separated from the LUVs using by size-exclusion chromatography (Sephadex G25-fine). Lipid concentration was determined by a colorimetric assay and size distribution of LUVs was evaluated by dynamic light scattering. For leakage, peptide assemblies (monomers, prefibrillar and fibrils) were prepared as described above and used at a final concentration of 50 μM. Calcein-LUVs were used at a final concentration of 500 μM. Fluorescence was monitored in sealed black-wall, clear-bottom 96 well non-binding surface plates with a total volume of 100 μL per well. Measurements were performed every 15 min using an excitation wavelength of 495 nm and emission at 517 nm. The control used to determine 100% leakage ($F_{max}$) was calcein-LUVs with 0.1% of Triton X-100. Dye leakage was reported using the following equation (Eq. 3):

$$\%membrane\ leakage = \frac{F - F_0}{F_{max} - F_0} \times 100 \qquad (3)$$

where $F_0$ is the fluorescence of the LUVs in absence of peptide. Data of at least four individual experiments performed with different lot of peptides were averaged and were expressed as the mean ± S.E.M.

**Confocal microscopy**. CHO-K1 ells were cultured in 8-well cell culture chamber at a density of 15,000 cells/well for 48 h before treatment. Alexa488-labelled oligomers or fibrils were added to the cell media and cells were incubated at 37 °C or 4 °C for 3 h. After incubation, cells were washed three times with PBS. Membrane was stained with 0.2 μg/mL of Cell mask and 0.1 μg/mL 4′,6-diamidino-2-phenylindole dihydrochloride (DAPI). Cells were then washed three times with PBS, fixed with 4% paraformaldehyde. Fluorescence was analysed with a confocal Nikon microscope equipped with a 60x oil immersion objective. Images were analysed using ImageJ software.

**Statistical analysis**. Statistical analyses were performed using the Student's t-test and statistically significance statistical difference (between IAPP and N21Q, otherwise stated) was established at $p < 0.001$ (*); $p < 0.0001$ (**); $p < 0.00001$ (***). Statistical analyses were carried out using GraphPad Prism 8.0 software.

**Reporting summary**. Further information on research design is available in the Nature Research Reporting Summary linked to this article.

## Data availability
The data supporting the findings of this manuscript are available from the corresponding authors upon reasonable request.

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

## Acknowledgements

This research was funded by the Natural Sciences and Engineering Research Council of Canada (NSERC), grant number RGPIN-2018-06209 to S.B. P.T.N., X.Z. and M.S. acknowledge fellowships from NSERC. The authors acknowledge Thierry Lefebvre and Elizabeth Godin for their technical assistance.

## Author contributions

P.T.N. and S.B. conceived and designed the overall study. P.T.N. performed the cell-based assays, fluorescence and CD spectroscopy, dot blot, membrane leakage and AFM. X.Z. performed the PXRD, FTIR, TEM analysis and the immunogold labelling. M.S., A.A. and I.M. designed, performed and analysed the NMR experiment. P.T.N. and S.B. analysed data and prepared the figures. P.T.N. and S.B. wrote the manuscript and all authors contributed to its revision.

## Competing interests

The authors declare no competing interests.
