## [Peer Review File · Communications Biology]

Reviewers' comments:

Reviewer #1 (Remarks to the Author):

This is an interesting article that focuses on an issue of great relevance in the comprehensive searching for identify-understand the diversity and polymorphism of amyloid assemblies and the complexity of quaternary structures associated with protein conformational diseases. However, it requires the contribution of results that certainly have and have not been considered to know:

It still remains unclear how N21Q fibrils and IAPP fibrils interact with each other to induce cross-seeding behaviors. It is necessary to discuss the effect of cross-sequence interactions between the IAPP fibrils, IAPP oligomers and N21Q fibrils on hybrid amyloid structures and aggregation kinetics.

Reviewer #2 (Remarks to the Author):

The paper by Nguyen et al describes fibrils of a N21D mutant of IAPP. The N21D fibrils apparently have cytotoxic properties, whereas for the WT IAPP toxicity has generally been associated with oligomers in the literature. The strength of the paper is that a battery of state-of-the-art techniques are used for the study, and the experiments appear to be well done. The weakness is with the conceptual design, and that much of the analysis and interpretation of the results is descriptive, relying on technical semantic flourishes and reiterations of amyloid dogma, rather than providing a mechanistic explanation. While there are probably enough variations of amyloid structures to launch a new journal, *Nature: Amyloid Polymorphism*, simply describing differences without providing new mechanistic insights is not very compelling. I also worry that for many of the studies insufficient experimental details are given to reproduce the results.

Major comments:

1. In two instances (lines 87 and 433) the authors state that the N21Q mutation adds a single methyl group. Unless my understanding of chemistry is way off, this must be "methylene" not "methyl"!

2. Relevance. An obvious selling point of the work is the relevance of amyloid to disease. But the N21Q mutation does not occur in nature, which makes me wonder about the generality or relevance of its particular polymorphism. How does it relate to the toxic worm-like aggregates when N21 is replaced by hydrophobic residues the authors report in reference 27? Would the worm-like aggregates be considered oligomers, fibrils, both, or something else? What was the rationale for choosing the "apparently trivial" N21Q mutant for study? For a 37-residue peptide, there are $20^{37} = 1048$ possible sequences. Is there a systematic approach for exploring sequence space, or is the MO to rely on the serendipitous finding of mutants with unusual properties?

3. Interpretation. Much of the very speculative discussion for the paper centers around the concept of oligomer-like fibrils (to add to monomers, oligomers, protofilaments, fibrils, and toxic worm-like aggregates, amongst others). One of the problems with the nomenclature is that fibrils are oligomers! The exposition gets very convoluted when talking about nucleation and cross-seeding reactions, since fibrils are formed through nucleation kinetics whereas oligomers are not. I think the authors use the "oligomer-like fibril" term to satisfy the current dogma that oligomers are bad and fibrils have limited effects on toxicity. But to me "oligomer-like fibrils" seems like saying a belly dancer is a human with worm-like properties because the belly dancer can writhe. And the convoluted word-play does not substitute for insight into what the "properties" that distinguish N21Q fibrils might be. For example:

3.1 In supporting Fig. S2 there appears to be a correlation between toxicity and twist repeat of the fibrils. Could this be what distinguishes N21Q fibrils? Rather than exploring this in more detail, the authors favor an explanation where the N21Q fibrils adopt a mysterious "property" of oligomers, since the prevailing literature says oligomers are likely to be toxic to cells. The problem is that the structural properties of oligomers are

not defined (or very poorly characterized) for IAPP, or any other type of amyloid! So whatever this property might be, it could be a moving target.

3.2 Have the authors considered that oligomers could be present together with the fibrils or that they might be dissociating from the fibrils (since the only isolation step seems to be a sedimentation step – line 146). If oligomers co-existed with the fibrils (through dissociation), and the equilibrium differed in N21Q compared to WT, it might explain the results on toxicity (Fig. 1C), kinetics (Fig. 4), lower ThT fluorescence (Fig. 4A), costaining with gold nanoparticles (Fig. 5), membrane disruption (Fig. 6), and A11 antibody reactivity (line 300).

3.3 I don't find the data on cross-seeding effects on toxicity in Fig. 7d all that convincing. At the highest seeding level (5%) IAPP fibrils achieve 60% viability whereas pure N21Q fibrils are 10% viable at 50 μ M concentrations in Fig. 1C. The 60% in Fig. 7d is in the ball park of the expected linear average (95% non-toxic IAPP, 5% N21Q at 10% viability) of the 70-90% viability with 2-5 μ M N21Q in Fig. 1C. In any case if the conversion of WT IAPP fibrils to N21Q-like fibrils were complete, the viability in Fig. 7D is considerably different from that in Fig. 1C. Of course, the results would depend on how accurately the concentrations of the peptides can be measured. How peptide concentrations were measured is not specified in the methods.

4. NMR

4.1 Line 229-235: In my opinion, the attribution of the change in CO shift of V32 to changes in hydrophobicity (which continues in the discussion) could be toned down or removed since it is very speculative. If anything, a change in hydrophobicity might be expected to affect side-chain resonances more than the backbone but besides the CO resonance, the sidechain peaks of V32 (beta 32 ppm, gMe# 19. ppm) look identical between the mutant and WT. The side-chain signals from the other residues also look unchanged.

4.2 The most striking thing about the NMR data in Fig. 3 is that there are pronounced changes in intensities (linewidths) of the CO resonances between WT and the mutant, suggesting some kind of dynamic effect. This might be worth a comment.

5. Lack of experimental details

5.1 Amyloid research has a history of oligomers, polymorphisms, and toxicity mechanisms that can only be achieved in one lab. While this might have the effect of shutting out other labs from funding and high-impact publications since they cannot reproduce the 'special sauce', it is probably not ideal for progress in the field. As such it seems important to specify exactly the methods used to generate the species reported. Since many in the literature have attributed IAPP cytotoxicity to oligomers, they have been much sought-after, albeit not that well characterized due to their low population and ephemeral nature. I was thus surprised that in this work oligomers were simply generated by incubating 150 μ M peptide for 15 min, whereas fibrils are formed with a longer incubation period of 48 h (Line 523 Methods). Is this standard procedure to make oligomers that could be used by other labs? What is the ratio of oligomers to monomers and fibrils under these conditions (I'm assuming that oligomers are not the only species present). Is the operational definition of an oligomer that it binds to the A11 antibody? Something else? Does monomer bind to this antibody as well (albeit with less avidity than the oligomer)? I'm raising these issues because they are not discussed in the paper, and no reference is given to the 15 min incubation method to prepare oligomers.

5.2 Line 588-599 states that fibrils for NMR were prepared at 750 μ M concentrations some 30-60 x higher concentrations than the 12.5 or 25 μ M used for kinetic studies (line 537). What was the reason for this and were any checks done to see that the fibril morphology did not change over this concentration range? How long were the 750 μ M fibrils incubated for?

5.3 Line 569. What were the growth conditions (concentration, time, etc) for the fibrils imaged by TEM?

5.4 Line 597-599. NMR referencing needs to be described better, or a reference should be given.

5.5 How were peptide concentrations measured? I don't think this is described anywhere in the paper. This is significant for the various protocols to produce oligomers & fibrils, their cytotoxicity (Fig. 1) as well as the seeding experiments (Fig. 7d).

Minor comments:

6. Line 128, Fig. 1A, right hand side Roder structure. I think the "N" and "C" terminal labels must be reversed. Look at the way the β -strand arrows are pointing!
7. Line 72 "... that cytotoxicity of could be independent". Something is missing in this sentence.
8. Fig. 1E. "Calcein" in the middle panels is not defined.
9. line 375 – "plays a critical in". Something missing.
10. line 375 thereabouts. I'm not entirely sure but I don't think seeding is entirely equivalent to secondary nucleation. Seeding can have to do with fibril elongation, rather than growth of new fibrils (nucleated by defects on pre-existing fibrils).
11. line 399. The "oligomer-like fibril" nomenclature is particularly inopportune here since fibrils are nucleated whereas oligomers are not.
12. lines 466-468. Attributing macroscopic differences in fibril morphology (by AFM) to variations in a single inter-protofilament H-bond is extremely speculative since there is no evidence presented in this paper that the N21Q mutation affects inter-protofilament H-bonds. Moreover, the H-bond in question for WT-IAPP is based on a 4.2 Å resolution cryo-EM data set, which makes me think the H-bond strength should be taken with a grain of salt even if it appears in a high IF-journal
13. lines 524-526. Labeled with what? Incubated for how long? What are the ratios of labeled: unlabeled about?
14. Lines 154-155: I find it very surprising that amyloid fibrils would unfold with a TM of 58 o C (even in 4 M urea). Has TEM been used to check that the fibrils are actually disrupted at high temperatures? Is the unfolding reversible? If not, it doesn't really fall in the realm of "thermodynamic stability" as stated in line 152.

Reviewer #3 (Remarks to the Author):

The work titled "Oligomer-like fibrils:....." by Nguyen et al. is a very extensive piece of work to show the role of single crucial residue/additional functional group in taking IAPP fibrillation from its own normal course of fibrillation, resulting in producing relatively very cytotoxic end-product, fibrils as per the authors. Although the data corroborate to support the conclusion drawn in the end, i.e. N21Q mutation in IAPP results in twisted thinner fibrils with higher cytotoxic propensity, minor points need to be addressed to make the manuscript more convincing to draw the conclusion, which are following:

1. For cytotoxicity assay, the cell is cultured in presence of either mature fibrils (24 hrs incubated monomers in fibrillation condition) or after certain time points of incubation like 5 hrs. Though both the proteins do not share same kinetics, it would have been advisable if authors, additionally, take respective incubated protein solutions from their respective phases, like log, exponential and plateau phases for cytotoxicity study. Or the authors may do additional cytotoxicity assay by incubating the cell with monomeric WT and mutant IAPP.

Somehow, from observing the represented data, it indicates that N21Q-IAPP fibrillation is kind of either delayed kinetics or the kinetics is trapped in oligomeric phase for longer time window (studied time window for the kinetics). Why so??? Because, when the author checked the cytotoxicity of 120 hrs incubated samples, it relatively loses the cytotoxic propensity, cell viability is 60% compared to 20% for 24 hrs incubated samples (SI fig. 2A). If it is true that the mutation stabilized (kind of) the fibrillation at oligomeric/protofibrils phase, it is justified to have higher cytotoxic propensity. As the authors have also proved and referenced in the manuscript, the flanking/dynamic residues (like SNFGAIL in this case) to internal axis of oligomer/protofibrils preferentially interact with cell membrane, leading to pore formation and cell death. Whereas, the interaction is sterically limited to superficial fibrils attachment to cell surface, as also shown in the manuscript.

2. IAPP fibrils seem double of the N21Q-IAPP fibrils height (AFM image) and the texture of the fibrils (mutated) corroborate with above assumption (point 1) that the mutated IAPP is stabilized in protofibrils phase (Zanetti, C I et al, PNAS, 1999, 96 (23), 1317509).

3. On contrary to the claim that (p20, I387) the N21Q conformation propagate to WT peptide, the IAPP fibrils, thus formed, has 40 % cytotoxicity only (cell viability = 60 % in presence of 5 % N21Q-IAPP seeds). Whereas N21Q fibrillar structure showed 80 % cytotoxicity. Hence, to validate

the statement, the authors need to quantify the F23 chemical shift for each seeded (at least for 3 percentage values) fibrils using ssNMR, as done for the normal fibrillation.

4. As stated in p12, l232-233 (...this is not related to alteration in secondary structure but may instead be a result of local increase in hydrophobicity), what if both the factors (structural rigidity for proximity to axil and increase in local effective hydrophobicity) are there?

Response to reviewers

Reviewer 1	
Referee comments	Responses
This is an interesting article that focuses on an issue of great relevance in the comprehensive searching for identify-understand the diversity and polymorphism of amyloid assemblies and the complexity of quaternary structures associated with protein conformational diseases.	We thank the reviewer 1 for his positive and constructive comments.
1. However, it requires the contribution of results that certainly have and have not been considered to know: It still remains unclear how N21Q fibrils and IAPP fibrils interact with each other to induce cross-seeding behaviors. It is necessary to discuss the effect of cross-sequence interactions between the IAPP fibrils, IAPP oligomers and N21Q fibrils on hybrid amyloid structures and aggregation kinetics.	1. We thank reviewer 1 for this comment and we agree with the reviewer. Accordingly, we have included additional data regarding kinetics of co-assembly to support the investigation of peptide cross-interactions. As requested by the reviewer, WT IAPP and N21Q were co-assembled and the kinetics of aggregation were monitored by ThT and ANS fluorescence. As observed in Supplementary Figure 10, gradual increase of the molar ratio of N21Q in IAPP self-assembly (from 1 to 10%) progressively diminished the lag-time, decreased the final ThT fluorescence signal and increased the final ANS fluorescence signal. These observations indicate that the presence of N21Q monomers influences the kinetics of IAPP self-assembly, supporting co-assembly of the monomeric peptides. The opposite effect was observed for the reverse experiment, i.e. when N21Q self-assembly occurs in presence of increasing molar ratio of WT IAPP; (i) augmentation of the lag phase, (ii) increase of the final ThT fluorescence, (iii) decrease of the final ANS fluorescence. Accordingly, the following sentence was incorporated in the Results (section Kinetics of self-assembly and time-resolved cytotoxicity): < Gradual augmentation of the molar ratio of N21Q into IAPP self-assembly (from 1 to 10%) progressively hastened nucleation and led to reduced final ThT fluorescence and increased final ANS fluorescence, while the opposite effect was observed for the reverse experiment, i.e. IAPP into N21Q assembly reaction (Supplementary Fig. 10). These observations suggest that IAPP and N21Q monomers co-assemble, leading to fibrils that progressively acquire the properties of their co-assembling counterpart. > Cross-seeding of IAPP with N21Q fibrils (Figure 7; Supplementary Figure 18) and cross-seeding of N21Q with WT IAPP fibrils (Supplementary Figure 20) revealed that the engendered assemblies progressively acquired the conformational and biological properties of their fibrillar counterparts. Thus, hybrid amyloid structures could be obtained by supplementing the self-assembly reaction with the corresponding fibril counterpart. Considering, the complexity and high heterogeneity of the oligomer mixtures, we did not performed relevant experiments regarding cross-sequence involving oligomers.
Reviewer 2	
Referee comments	Responses
The paper by Nguyen et al describes fibrils of a N21D mutant of IAPP. The N21D fibrils apparently have cytotoxic properties, whereas for the WT IAPP toxicity has generally been associated with oligomers in the literature. The strength of the paper is that a battery of state-	We thank reviewer 2 for his very exhaustive review and for his constructive comments. We agree with reviewer 2 regarding the variations and high polymorphism of amyloid assemblies available in the literature. This high polymorphism as well as the transient/dynamic nature of the off- and on-pathway intermediates make it very challenging to define the structural motif(s)

of-the-art techniques are used for the study, and the experiments appear to be well done. The weakness is with the conceptual design, and that much of the analysis and interpretation of the results is descriptive, relying on technical semantic flourishes and reiterations of amyloid dogma, rather than providing a mechanistic explanation. While there are probably enough variations of amyloid structures to launch a new journal, Nature: Amyloid Polymorphism, simply describing differences without providing new mechanistic insights is not very compelling. I also worry that for many of the studies insufficient experimental details are given to reproduce the results. Major comments:	associated with cytotoxicity. Herein, we showed that the ‘oligomer-like’ conformation, which we mainly defined by the fact that it is specifically recognized by the A11 antibody, can be preserved upon fibril elongation and can propagate to WT peptide. We consider that the identification of such fibrils bearing an ‘oligomer-epitope’ is important for the field, since it opens to the possibility of structurally defining the molecular determinants of amyloidogenesis-mediated cell death. We apologize for any insufficient experimental details provided in the first version of the manuscript. In the revised version, we have included the necessary details to reproduce systematically all experiments of this study (see below and in the modified manuscript).
1. In two instances (lines 87 and 433) the authors state that the N21Q mutation adds a single methyl group. Unless my understanding of chemistry is way off, this must be “methylene” not “methyl”!	1. We thank the reviewer for noticing this mistake. This has been modified accordingly.
2. Relevance. An obvious selling point of the work is the relevance of amyloid to disease. But the N21Q mutation does not occur in nature, which makes me wonder about the generality or relevance of its particular polymorphism. How does it relate to the toxic worm-like aggregates when N21 is replaced by hydrophobic residues the authors report in reference 27? Would the worm-like aggregates be considered oligomers, fibrils, both, or something else? What was the rationale for choosing the “apparently trivial” N21Q mutant for study? For a 37-residue peptide, there are $20^{37} = 1048$ possible sequences. Is there a systematic approach for exploring sequence space, or is the MO to rely on the serendipitous finding of mutants with unusual properties?	2. The modus operandi is not simply to rely on a serendipitous finding of mutants. As mentioned in the introduction, in a previous structure-self-assembly study (Godin, JBC, 2019), we reported that the N21 residue constitutes a molecular hinge that modulates IAPP conformational transition associated with self-assembly and toxicity. In the course of this investigation, we observed that the N21Q substitution led to the formation of well-defined fibrillar assemblies, which have high toxicity and are recognized by the A11 oligomers, which is a novel and important finding. Thus, the rationale for choosing the N21Q substitution is because Asn-21 is critical to control the conversion from cytotoxic proteospecies into non-toxic fibrils. Of course, we could not predict that the N21Q substitution would lock the fibril into a transmissible structure that is recognized by the A11 antibody, and we recognize this aspect in the introduction. The relevance of this paper is based on the finding that this ‘conformational motif’ recognized by the A11 antibody, which is usually not nucleating competent, can propagate to naïve peptide. Besides, to avoid any possible confusion, reference to the ‘worm-like aggregates’ was removed.
3. Interpretation. Much of the very speculative discussion for the paper centers around the concept of oligomer-like fibrils (to add to monomers, oligomers, protofilaments, fibrils, and toxic worm-like aggregates, amongst others). One of the problems with the nomenclature is that fibrils are oligomers! The exposition gets very convoluted when talking about nucleation and cross-seeding reactions, since fibrils are formed through nucleation kinetics whereas oligomers are not. I think the authors use the “oligomer-like fibril” term to satisfy the current dogma that oligomers are bad and fibrils have limited effects on toxicity. But to me “oligomer-like fibrils” seems like saying a belly dancer is a human with worm-like properties because the belly dancer can writhe. And the convoluted word-play does not	3. The term ‘oligomer-like fibrils’ refers to assemblies with a fibrillar mesoscopic morphology (linear, long and unbranched filaments) and that, in contrast to the vast majority of the prototypical amyloid fibrils, are highly cytotoxic, are recognized by the A11 antibody and display other typical characteristics of oligomers (low ThT binding, persistent capacity to disrupt lipid membranes). The ability of these fibrils to seed new populations of fibrils, that acquired these ‘oligomer-like’ properties, is key. The proposed nomenclature was selected because the term ‘fibril-like oligomers’ has been employed in the literature, including the fibrillar oligomers (TABFOs) of the Eisenberg group, which are oligomers that share structural similarities with mature amyloid fibrils, i.e. the opposite of the supramolecular structure presented in the present study.

substitute for insight into what the “properties” that distinguish N21Q fibrils might be. For example: 3.1. In supporting Fig. S2 there appears to be a correlation between toxicity and twist repeat of the fibrils. Could this be what distinguishes N21Q fibrils? Rather than exploring this in more detail, the authors favor an explanation where the N21Q fibrils adopt a mysterious “property” of oligomers, since the prevailing literature says oligomers are likely to be toxic to cells. The problem is that the structural properties of oligomers are not defined (or very poorly characterized) for IAPP, or any other type of amyloid! So whatever this property might be, it could be a moving target. 3.2. Have the authors considered that oligomers could be present together with the fibrils or that they might be dissociating from the fibrils (since the only isolation step seems to be a sedimentation step – line 146). If oligomers co-existed with the fibrils (through dissociation), and the equilibrium differed in N21Q compared to WT, it might explain the results on toxicity (Fig. 1C), kinetics (Fig. 4), lower ThT fluorescence (Fig. 4A), costaining with gold nanoparticles (Fig. 5), membrane disruption (Fig. 6), and A11 antibody reactivity (line 300).	3.1. As noticed by the reviewer, the quantification of over 3000 fibrils per peptide using AFM images revealed that the prevalence of twisted fibrils in the N21Q samples was significantly higher compared to IAPP fibrils. Indeed, as mentioned in the paper, this could be associated with the observed higher toxicity of N21Q fibrils. Besides, the average height of the N21Q fibrils is approximately half of the average height of the WT IAPP fibrils, i.e. 3.12 ± 1.24 nm vs 5.98 ± 2.62 nm, as measured by AFM. Thus, the N21Q substitution could lock the self-assembly process into fibrils/protofibrils that expose lattice of the SNFGAIL domain. We fully agree with the reviewer that the structural properties of oligomers are poorly characterized in the literature, for IAPP and other amyloidogenic proteins. In this paper, we show that the N21Q fibrils have the ability of propagating to new populations of fibrils that display the oligomer A11 conformational epitope, which will ultimately support the structural characterization of oligomers. 3.2. We thank the reviewer for this constructive and important comment. We did consider that oligomers could be present within the N21Q fibril mixture, or that oligomers could dissociate from the mature N21Q fibrils. We investigated if the unexpected toxicity of N21Q fibrils could result from the presence of soluble oligomers associated with an incomplete fibrillization and/or dissociation of oligomers from the fibrils. Experimental evidences showing that this is not likely the case, include; 1- Peptides were assembled for up to 5 days and cytotoxicity of the aggregation mixtures was periodically evaluated. Prolonging self-assembly did not modify the divergent toxicity observed between WT and N21Q fibrils (Supplementary Fig. 2), suggesting that toxicity is likely not associated with an incomplete fibrillization. 2- Fibrils were isolated by multiple centrifugation/sonication to remove any remaining soluble species that could cause cell death. Fibrils were subjected, or not, to a 30 min sonication, before being centrifuged at 35 000 g for 45 min. Pellets were washed, sonicated again or not, and centrifuged a second time. First supernatants and final pellets were analyzed by transmission electron microscopy (TEM) and cytotoxicity was evaluated. N21Q fibrils isolated in the final pellet, with or without sonication, remained toxic to pancreatic cells (Supplementary Fig. 3). 3- As fibrils can disassemble and release oligomers, we evaluated the thermodynamic stability by thermal denaturation monitored by circular dichroism (CD) spectroscopy. According to the β-sheet signal at 218 nm, IAPP and N21Q fibrils in presence of 4 M urea showed a similar thermal unfolding midpoint (Supplementary Fig. 4). 4- Stability against proteolysis was assessed by subjecting the assemblies to proteinase K (PK) digestion. N21Q and WT fibrils showed an equivalent stability against PK proteolysis, suggesting a similar degree of fibril compactness and undetectable dissociation of soluble oligomer species (Supplementary Fig. 5). 5- The kinetics of lipid membrane disruption and caspase 3 activation for IAPP oligomers and isolated N21Q fibrils (both at 50 μM, monomer equivalent) are the same (Figure 6), which most likely rule out the fact that toxicity would be associated with oligomer dissociation, as the concentration of oligomers in the N21Q fibril solution would be minimal compared to the effective concentration heterogeneous oligomer preparation. Together, these observations strongly suggest that the cytotoxicity of N21Q fibrils is mediated by defined N21Q fibrils and is not associated with the presence of soluble oligomers or dissociation of oligomers from fibrils.
---	---

3.3. I don't find the data on cross-seeding effects on toxicity in Fig. 7d all that convincing. At the highest seeding level (5%) IAPP fibrils achieve 60% viability whereas pure N21Q fibrils are 10% viable at 50 uM concentrations in Fig. 1C. The 60% in Fig. 7d is in the ball park of the expected linear average (95% non-toxic IAPP, 5% N21Q at 10% viability) of the 70-90% viability with 2-5 uM N21Q in Fig. 1C. In any case if the conversion of WT IAPP fibrils to N21Q-like fibrils were complete, the viability in Fig. 7D is considerably different from that in Fig. 1C. Of course, the results would depend on how accurately the concentrations of the peptides can be measured. How peptide concentrations were measured is not specified in the methods.	3.3. We agree with the reviewer. IAPP fibrils seeded with 2.5% and 5% N21Q fibrils led to 60% of cell viability, in contrast to 10 to 20% for the equivalent homogenous N21Q fibril preparation. This difference is likely ascribed to the formation of a mixed population of protofilaments/fibrils upon seeding with N21Q seeds. As suggested by the reviewer 3 (see below), we site-specifically labelled IAPP (13C, 15N) at position Phe23 and the self-assembly of F23 (13C, 15N) labelled IAPP was performed in presence of unlabeled N21Q seeds (0.5%, 2.5% and 5%). After self-assembly, the fibrils were isolated and analyzed by SS-NMR (CP-DARR), as performed in Figure 3. The ssNMR experiments highlighted the heterogeneity of the seeded fibril population, likely explaining the 60% of cell viability (compared to 10 to 20% at equivalent concentration of N21Q fibrils). A new panel was incorporated in Figure 7 (SS-NMR). Regarding how peptide concentration was measured, the concentrations are basically determined by weight (using a high-precision ultra-microbalance) and this could, in theory, be associated with large error. However, according to our sample preparation protocol, we weight at least 3 to 6 mg of lyophilized/monomerized peptide on an ultra-microbalance after the first step of the monomerization protocol with HFIP. Thus, we are accurate regarding the weight measured. Moreover, we are validating peptide concentration by measuring absorbance at 280 nm using a theoretical molar extinction coefficient of $1490 \text{ M}^{-1} \text{ cm}^{-1}$. Because of the absence of a Trp residue in IAPP sequence, using absorbance at 280 nm can lead to important variation. Accordingly, we are also validating the concentration of each peptide batch by measuring the area under curve (229 nm) of a RP-HPLC run of a monomerized/lyophilized peptide aliquot. The following information was included in Methods (Peptide synthesis, purification and sample preparation): < The resulting peptide powder was weighted using an ultra-microbalance, solubilized for a second time in HFIP to 1 mg/ml, sonicated for 30 min and lyophilized again. Peptide concentrations were validated by measuring absorbance at 280 nm using a theoretical molar extinction coefficient of $1490 \text{ M}^{-1} \text{ cm}^{-1}$ and by measuring the area under curve at 229 nm by RP-HPLC. Monomerized dried samples were kept dried at -80°C until used, but not for longer than 4 weeks. >
4. NMR: 4.1. Line 229-235: In my opinion, the attribution of the change in CO shift of V32 to changes in hydrophobicity (which continues in the discussion) could be toned down or removed since it is very speculative. If anything, a change in hydrophobicity might be expected to affect side-chain resonances more than the backbone but besides the CO resonance, the sidechain peaks of V32 (beta 32 ppm, gMe# 19. ppm) look identical between the mutant and WT. The side-chain signals from the other residues also look unchanged.	4.1. We agree with the reviewer that this affirmation regarding V32 was somewhat speculative. As suggested, we tuned down this hypothesis. This hypothesis is only mentioned in the Results: < Considering the lack of changes in the $\text{C}\alpha$ and $\text{C}\beta$ secondary shifts of F23, this is not likely related to a major alteration in secondary structure, but may instead be a result of a local increase in hydrophobicity⁴³ and/or from altered hydrogen-bond interactions⁴⁴. Change in V32 CO secondary shifts, from negative value towards zero, could suggest the possibility that a similar effect was occurring around the C-terminal region. > 43. Mallamace, D., Fazio, E., Mallamace, F. & Corsaro, C. The Role of Hydrogen Bonding in the Folding/Unfolding Process of Hydrated Lysozyme: A Review of Recent NMR and FTIR Results. Int J Mol Sci 19, 3825, doi:10.3390/ijms19123825 (2018). 44. Wei, Y., Lee, D. K. & Ramamoorthy, A. Solid-state (13)C NMR chemical shift anisotropy tensors of polypeptides. Journal of the American Chemical Society 123, 6118-6126, doi:10.1021/ja010145I (2001).
4.2. The most striking thing about the NMR data in Fig. 3 is that there are pronounced changes in intensities (linewidths) of the CO resonances between WT and the mutant, suggesting some kind of dynamic effect. This might be worth a comment.	4.2. We thank the reviewer for noticing it. In fact, the changes in intensity are probably not indicating of different dynamics, but, from our opinion, are simply a result of sample quantity. This is supported by the fact that the signal/noise is overall somewhat lower in IAPP than N21Q. The signal obtained from WT fibrils was lower than that of N21Q fibrils due to the differences in recovery after lyophilization. For IAPP fibrils, approximately

	50% of the starting mass was recovered, whereas for N21Q fibrils, almost 80% of the starting mass was recovered. In retrospective, maybe we should have adjusted the spectral intensities to make them equal, although we wanted to conserve equivalent spectral intensities. Accordingly, we added the following sentence in Methods (Solid-state nuclear magnetic resonance): < For IAPP fibrils, approximately 50% of the starting mass was recovered, whereas for N21Q fibrils, almost 80% of the starting mass was recovered. >
5. Lack of experimental details 5.1 Amyloid research has a history of oligomers, polymorphisms, and toxicity mechanisms that can only be achieved in one lab. While this might have the effect of shutting out other labs from funding and high-impact publications since they cannot reproduce the ‘special sauce’, it is probably not ideal for progress in the field. As such it seems important to specify exactly the methods used to generate the species reported. Since many in the literature have attributed IAPP cytotoxicity to oligomers, they have been much sought-after, albeit not that well characterized due to their low population and ephemeral nature. I was thus surprised that in this work oligomers were simply generated by incubating 150 μM peptide for 15 min, whereas fibrils are formed with a longer incubation period of 48 h (Line 523 Methods). Is this standard procedure to make oligomers that could be used by other labs? What is the ratio of oligomers to monomers and fibrils under these conditions (I’m assuming that oligomers are not the only species present). Is the operational definition of an oligomer that it binds to the A11 antibody? Something else? Does monomer bind to this antibody as well (albeit with less avidity than the oligomer)? I’m raising these issues because they are not discussed in the paper, and no reference is given to the 15 min incubation method to prepare oligomers.	5.1. We fully agree with reviewer 2 regarding the fact that research on amyloid protein assemblies is particularly associated with lack of reproducibility between studies/laboratories. In the present study, we established thorough, albeit straightforward, procedures to accurately and easily reproduce the preparation of the different assemblies. The monomerization protocol (two successive HFIP treatment, sonication and filtration) is particularly important. We and other research groups have previously described this protocol, which is standard for IAPP. The buffer composition (20 mM Tris-HCl buffer, pH 7.4) is also important for the kinetics of self-assembly and this buffer is used by the main research groups working on IAPP, including the group of Daniel Raleigh at Stony Brook. For fibril formation, the 48 h incubation is standard for IAPP. In fact, under quiescent conditions and at 150 μM, mature fibrils are formed after 12 h (see panel h, Figure 4). The equilibrium phase under these conditions is reached before 24 h. For the preparation of the oligomer mixture, it is definitively less well defined in the literature. We have shown that IAPP oligomerizes rapidly under neutral pH (Quittot, ACS Bioconjugate Chemistry, 2018). In the present study, we showed by PICUP (Supplementary Fig. 12), that these conditions (150 μM, 15 mins, and quiescent conditions) led to a heterogeneous mixture of small oligomers (dimer to hexamer/heptamer) and monomers. It is difficult to estimate the ratio of oligomers to monomers, because of the heterogeneity of mixtures and we are not fully confident of estimating these ratio from the silver stained SDS-PAGE of cross-linked oligomers. At a concentration of 150 μM, small protofibrils are already observable after 45 to 60 mins incubation by TEM imaging. We have now included the reference to our ACS Bioconjugate Chemistry in the Methods section. 45. Quittot, N., Sebastiao, M., Al-Halifa, S. & Bourgault, S. Kinetic and Conformational Insights into Islet Amyloid Polypeptide Self-Assembly Using a Biarsenical Fluorogenic Probe. Bioconjugate Chemistry 29, 517-527, doi:10.1021/acs.bioconjchem.7b00827 (2018). Yes, the ‘operational definition’ of an oligomer could be associated to recognition by the A11 antibody. As requested by the reviewer, we performed dot blot analysis with freshly monomerized dissolved peptide (mostly monomeric) and we did not observe any recognition by the A11 antibody (Supplementary Fig. 13). The following sentence was incorporated into the text (Results): < No binding of the A11 antibody to freshly dissolved monomerized peptides was observed by dot blot analysis (Supplementary Fig. 13). >
5.2 Line 588-599 states that fibrils for NMR were prepared at 750 μM concentrations some 30-60 x higher concentrations than the 12.5 or 25 μM used for kinetic studies (line 537). What was the reason for this and were any checks done to see that the fibril morphology did not change over this concentration range? How long were the 750 μM fibrils incubated for?	5.2. For the SS-NMR experiments, fibrils were prepared at high concentration considering the amount of fibrils needed. The concentration is 375 μM and not 750 μM. We apologize for this mistake and we have corrected this information. Of course, we validated that the distinctive biophysical, (supra)structural and biological properties of the N21Q fibrils are retained. As observed in Supplementary Fig. 7, N21Q fibrils assembled at 375 μM were thinner compared to IAPP fibrils, showed a concentration-dependent toxicity, low ThT-binding and high surface hydrophobicity. The 375 μM concentration is 2.5x higher to the standard 150 μM working concentration. The 375 μM peptide for the SS-NMR experiments were incubated for 48 h. This information has been included in the revised manuscript:

	< Fibrils assembled at 375 μM for 48 h were concentrated by ultracentrifugation at 100 000g for 45 min at 4 °C. >
5.3 Line 569. What were the growth conditions (concentration, time, etc) for the fibrils imaged by TEM?	5.3 The concentration was 150 μM. For clarity, we added this information in the text. Please note that the following information is available in Methods (Preparation of oligomers and fibrils) : < Fibrils were prepared by incubating freshly dissolved monomerized peptides under quiescent conditions at 150 μM in 20 mM Tris-HCl buffer, pH 7.4 for 48 h, unless otherwise stipulated. >
5.4 Line 597-599. NMR referencing needs to be described better, or a reference should be given.	5.4 We thank the reviewer for this comment. As requested we added a reference and we now better described the referencing in the Methods (Solid-state nuclear magnetic resonance) : < All experiments were recorded at 277 K and spectra were externally referenced using adamantane, setting the ¹³C methylene (CH₂) signal to 38.48 ppm, relative to tetramethylsilane (TMS).⁷³ > 73. Morcombe, C. R. & Zilm, K. W. Chemical shift referencing in MAS solid state NMR. J Magn Reson 162, 479-486, doi:10.1016/s1090-7807(03)00082-x (2003).
5.5 How were peptide concentrations measured? I don't think this is described anywhere in the paper. This is significant for the various protocols to produce oligomers & fibrils, their cytotoxicity (Fig. 1) as well as the seeding experiments (Fig. 7d).	5.5 Peptide concentrations are determined by weight (using a high-precision ultra-microbalance). This could, in theory, be associated with large error. However, according to our sample preparation protocol, we weight 3 to 6 mg of lyophilized/monomerized peptide on an ultra-microbalance after the first step of the monomerization protocol with HFIP. Thus, we are accurate regarding the weight measured. Moreover, we are validating peptide concentration by measuring absorbance at 280 nm using a theoretical molar extinction coefficient of 1490 M ⁻¹ cm ⁻¹ . Because of the absence of a Trp residue in IAPP sequence, using absorbance at 280 nm can lead to important variation. Accordingly, we are also validating the concentration of each peptide batch by measuring the area under curve (229 nm) of a RP-HPLC run. The following information was included in the Methods (Peptide synthesis, purification and sample preparation) : < The resulting peptide powder was weighted using an ultra-microbalance, solubilized for a second time in HFIP to 1 mg/ml, sonicated for 30 min and lyophilized again. Peptide concentrations were validated by measuring absorbance at 280 nm using a theoretical molar extinction coefficient of 1490 M⁻¹ cm⁻¹ and by measuring the area under curve at 229 nm by RP-HPLC. Monomerized dried samples were kept dried at -80°C until used, but not for longer than 4 weeks. >
Minor comments: 6. Line 128, Fig. 1A, right hand side Roder structure. I think the "N" and "C" terminal labels must be reversed. Look at the way the β-strand arrows are pointing!	6. We thank the reviewer for noticing it. This has been corrected.
7. Line 72 "... that cytotoxicity of could be independent". Something is missing in this sentence.	7. We thank the reviewer for noticing it. The sentence was changed to: <...it has also been proposed that cytotoxicity could be independent of a given structure.²⁴ >
8. Fig. 1E. "Calcein" in the middle panels is not defined.	8. Calcein refers to calcein-AM, which is used to show living cells by their intracellular esterase activity. Thus, to be more precise, on Figure 1E calcein was replaced by Calcein-AM. In addition, the following information were added in Methods (Cell viability) : < For Live/Dead assays, cells were plated in 12-well plates at density of 150 000 cells/well for 48 h before treatment. Analyses were performed by the addition of the reagent solution (4 μM of ethidium homodimer-1; 2 μM of calcein-AM) and after 45 min incubation, cells were imaged by fluorescent microscopy. >
9. line 375 – "plays a critical in". Something	9. We thank the reviewer for noticing it. Modification has been applied:

missing.	<...plays a critical role in amyloid formation.⁵²>
10. line 375 thereabouts. I'm not entirely sure but I don't think seeding is entirely equivalent to secondary nucleation. Seeding can have to do with fibril elongation, rather than growth of new fibrils (nucleated by defects on pre-existing fibrils).	10. We agree with reviewer 2 that seeding is not totally equivalent to secondary nucleation. To be more accurate, we modified the text to: <It is now known that the conformation of an aggregate can propagate to a naïve polypeptide building block, whose assemblies will ultimately acquire the properties of the pre-formed aggregates. This could occur through secondary nucleation, which corresponds to the catalyzed nucleation of monomers or oligomers by pre-assembled fibrils⁵², and/or through the elongation of pre-existing fibrillar seeds by the addition of monomers/oligomers to the growing end. Accordingly, we evaluated if the properties of the N21Q fibrils can be propagated to WT IAPP by supplementing IAPP self-assembly reaction with pre-assembled N21Q fibrils.>
11. line 399. The "oligomer-like fibril" nomenclature is particularly inopportune here since fibrils are nucleated whereas oligomers are not.	11. We agree with the reviewer that oligomers are not nucleated species per se , whereas fibrils are nucleated. However, the term 'oligomer-like fibril' refers to fibrils (nucleated) that have oligomer characteristics, such as recognition by A11 antibody, and not to oligomers that have fibril characteristics. In fact, what is interesting with these N21Q fibrils is that the 'A11 conformational epitope' associated with toxicity can propagate to naïve peptide, which is a new aspect for amyloid-associated assemblies. Nonetheless, to avoid any possible confusion, we modified this sentence to: <Overall, these results indicate that the conformation of N21Q fibrils that is recognized by the A11 antibody is transmissible to the WT peptide, which implies that the cytotoxic motif constitutes an organized supramolecular architecture.>
12. lines 466-468. Attributing macroscopic differences in fibril morphology (by AFM) to variations in a single inter-protofilament H-bond is extremely speculative since there is no evidence presented in this paper that the N21Q mutation affects inter-protofilament H-bonds. Moreover, the H-bond in question for WT-IAPP is based on a 4.2 Å resolution cryo-EM data set, which makes me think the H-bond strength should be taken with a grain of salt even if it appears in a high IF-journal	12. We agree with the reviewer that this hypothesis was speculative. Accordingly, we modify the text to: <As observed by AFM, the N-to-Q substitution affects twisting and/or wrapping of protofilaments, and the height and stiffness of the resulting fibrils, which could be related to changes in the strength and/or the positioning of hydrogen bonds involving residue-21 side chain.>
13. lines 524-526. Labeled with what? Incubated for how long? What are the ratios of labeled: unlabeled about?	13. We apologize for the lack of details. We modified the text to: <Fluorescently labelled peptide oligomers and fibrils were prepared by incubating non-labelled peptides with Alexa Fluor 488-labelled peptides (99.5:0.5, molar ratio) under quiescent conditions for 48 h at a final concentration of 150 µM.>
14. Lines 154-155: I find it very surprising that amyloid fibrils would unfold with a TM of 58 oC (even in 4 M urea). Has TEM been used to check that the fibrils are actually disrupted at high temperatures? Is the unfolding reversible? If not, it doesn't really fall in the realm of "thermodynamic stability" as stated in line 152.	14. We thank the reviewer for this comment. As recommended, we checked that the high temperature and 4 M urea treatment actually disrupt the fibrils by AFM. As observed in Supplementary Fig. 4C , this treatment led to a sharp decrease of the amount of fibrils, although some residual short fibrils could be observed on the mica. This 'unfolding' was not reversible in presence of 4 M urea and accordingly, the term 'thermodynamic stability' was removed.
Reviewer 3	
Referee comments	Responses
The work titled "Oligomer-like fibrils:....." by Nguyen et al. is a very extensive piece of work to show the role of single crucial residue/additional functional group in taking	We thank the reviewer for this positive comment and for his constructive suggestions.

IAPP fibrillation from its own normal course of fibrillation, resulting in producing relatively very cytotoxic end-product, fibrils as per the authors. Although the data corroborate to support the conclusion drawn in the end, i.e. N21Q mutation in IAPP results in twisted thinner fibrils with higher cytotoxic propensity, minor points need to be addressed to make the manuscript more convincing to draw the conclusion, which are following:	
1. For cytotoxicity assay, the cell is cultured in presence of either mature fibrils (24 hrs incubated monomers in fibrillation condition) or after certain time points of incubation like 5 hrs. Though both the proteins do not share same kinetics, it would have been advisable if authors, additionally, take respective incubated protein solutions from their respective phases, like log, exponential and plateau phases for cytotoxicity study. Or the authors may do additional cytotoxicity assay by incubating the cell with monomeric WT and mutant IAPP. Somehow, from observing the represented data, it indicates that N21Q-IAPP fibrillation is kind of either delayed kinetics or the kinetics is trapped in oligomeric phase for longer time window (studied time window for the kinetics). Why so??? Because, when the author checked the cytotoxicity of 120 hrs incubated samples, it relatively loses the cytotoxic propensity, cell viability is 60% compared to 20% for 24 hrs incubated samples (SI fig. 2A). If it is true that the mutation stabilized (kind of) the fibrillation at oligomeric/protofibrils phase, it is justified to have higher cytotoxic propensity. As the authors have also proved and referenced in the manuscript, the flanking/dynamic residues (like SNFGAIL in this case) to internal axis of oligomer/protofibrils preferentially interact with cell membrane, leading to pore formation and cell death. Whereas, the interaction is sterically limited to superficial fibrils attachment to cell surface, as also shown in the manuscript.	1. We thank the reviewer for this comment and we apologize for not being clear. In fact, as observed by ThT, ANS and FIAsH fluorescence, the N21Q substitution accelerates fibrillization of IAPP (Fig. 4, Fig.S9) and the assembled fibrils conserve high toxicity and oligomer-like properties (i.e. recognition by the A11 antibody, exposure of Phe23, capacity to perturb lipid membranes). The lower toxicity for N21Q fibrils observed after extended incubation (i.e., 120 h; SI Fig 2A, Fig. 4D) is likely associated with association/clumping of the protofilaments/fibrils into larger assemblies, leading to lower interaction with the cell membrane. To address the reviewer's comment, we modify Figure 4 to include additional data regarding the characterization of the proteospecies generated during the time-resolved analysis of cytotoxicity. We now included, time-resolved ThT, ANS and TEM analysis of the proteospecies that were generated after 0 h, 2 h, 4 h, 6 h, 12 h, 24 h, 48 h, 72 h, 96 h and 120 h. Particularly, well-defined fibrils could be observed by TEM after 2 h incubation (Fig. 4h) of the N21Q derivative, a time period associated with high toxicity. Accordingly, the following text was modified in the legend of Figure 4: <d Time-resolved cytotoxicity of proteospecies evaluated by measuring the metabolic activity of INS-1E upon 5 h incubation with 50 μM pre-assembled peptides. e-h Time-resolved self-assembly of IAPP and N21Q monitored by (e) CD spectroscopy, (f) ThT fluorescence, (g) ANS fluorescence and (h) TEM. d-h Freshly dissolved monomerized peptides were incubated under quiescent conditions at 150 μM in 20 mM Tris-HCl buffer, pH 7.4 and after the indicated time of self-assembly, the aggregation mixture was evaluated for cell toxicity and characterized.> The following text was added to the manuscript: <For time-resolved toxicity, peptides were incubated at 150 μM and after different incubation periods self-assembly was monitored by CD spectroscopy, ThT and ANS fluorescence, and TEM, and the toxicity of the proteospecies was evaluated by monitoring the viability of INS-1E cells after 5 h incubation with 50 μM pre-assembled proteospecies. > <Moreover, ThT and ANS fluorescence reached the plateau after 6 h incubation for N21Q and well-defined (proto)fibrils were observed by TEM imaging after only 2 h incubation (Fig. 4f, 4g, 4h), i.e. when cytotoxicity remains very high. In sharp contrast for IAPP, ThT and ANS signal reached the plateau after 12 to 24 h incubation and defined fibrils could be observed by TEM after 6 to 12 h. Taken together, these data suggest that N21Q substitution hastens IAPP self-assembly into fibrillar assemblies, and these N21Q fibrils conserve high toxicity.>
2. IAPP fibrils seem double of the N21Q-IAPP fibrils height (AFM image) and the texture of the fibrils (mutated) corroborate with above assumption (point 1) that the mutated IAPP is stabilized in protofibrils phase (Zanetti, C I et al, PNAS, 1999, 96 (23), 1317509).	As noted by the reviewer, the height of the N21Q fibrils is roughly half of the height of the WT IAPP fibrils, i.e. 3.12 ± 1.24 nm vs 5.98 ± 2.62 nm. In addition, as also mentioned by the reviewer, N21Q fibrils show a high proportion of twisted fibrils. These observations corroborate with the hypothesis that the N21Q substitution locks the self-assembly process into fibrils that exposed lattice of the SNFGAIL, which could be related to protofibrils. Accordingly, we introduced/modified the following text in the discussion:

	< Although both fibril preparations showed high polymorphism, the prevalence of twisted fibrils was significantly more important for N21Q. Moreover, as observed by AFM imaging, the height of N21Q filaments was approximately half of the height of IAPP fibrils. Thus, the N21Q substitution likely blocks the self-assembly process into twisted protofibrils by precluding their packaging in fibrils, as previously observed for the amyloid self-assembly of the immunoglobulin light chain⁶⁶> The following reference was added: 66. Ionescu-Zanetti, C. et al. Monitoring the assembly of Ig light-chain amyloid fibrils by atomic force microscopy. Proceedings of the National Academy of Sciences of the United States of America 96, 13175-13179, doi:10.1073/pnas.96.23.13175 (1999).
3. On contrary to the claim that (p20, l387) the N21Q conformation propagate to WT peptide, the IAPP fibrils, thus formed, has 40 % cytotoxicity only (cell viability = 60 % in presence of 5 % N21Q-IAPP seeds). Whereas N21Q fibrillar structure showed 80 % cytotoxicity. Hence, to validate the statement, the authors need to quantify the F23 chemical shift for each seeded (at least for 3 percentage values) fibrils using ssNMR, as done for the normal fibrillation.	We fully agree with the reviewer and we thank him for his clever suggestion. Indeed, IAPP fibrils seeded with 2.5% and 5% N21Q fibrils led to 60% of cell viability, in contrast to 10 to 20% for the equivalent homogenous N21Q fibril preparation. This difference is likely ascribed to the formation of a mixed population of protofilaments/fibrils upon seeding with N21Q seeds. As suggested by the reviewer, we site-specifically labelled IAPP (¹³C, ¹⁵N) at position Phe23 and the self-assembly of labelled IAPP was seeded with unlabeled N21Q seeds, (0.5%, 2.5% and 5%). After self-assembly, the fibrils were isolated and analyzed by SS-NMR (CP-DARR), as performed in Figure 3. Results revealed that the CO chemical shift of F23 progressively evolves from 171.3 ppm for unseeded IAPP fibrils to 174.1 ppm for IAPP fibril seeded with 5% N21Q seeds, which is closely related to the chemical shift of N21Q filaments (labelled at F23). Particularly, the ssNMR experiments highlighted the heterogeneity of the seeded fibril population, likely explaining, the 60% of cell viability. A new panel was incorporated in Figure 7 (SS-NMR). The following text was modified in the legend of Figure 7: <(c) SS-NMR (CP-DARR) of F23 labelled IAPP...> The following text was added to the manuscript: < Next, IAPP labelled at position F23 (¹³C, ¹⁵N) was incubated with unlabelled N21Q fibrils and the CO chemical shift of the seeded fibrils was compared with homogenous labelled IAPP and N21Q fibrils. The carbonyl region of the CP-DARR spectra revealed that the CO chemical shift of F23 was progressively deshielded as the molar ratio of N21Q seeds increased (Fig. 7c), indicative of a change in the environment of F23 IAPP towards the N21Q conformation. >
4. As stated in p12, l232-233 (...this is not related to alteration in secondary structure but may instead be a result of local increase in hydrophobicity), what if both the factors (structural rigidity for proximity to axil and increase in local effective hydrophobicity) are there?	We thank the reviewer for this comment and the text was modified accordingly: < Considering the lack of changes in the Cα and Cβ secondary shifts of F23, this is not likely related to a major alteration in secondary structure, but may instead be a result of a local increase in hydrophobicity⁴³ and/or from altered hydrogen-bond interactions⁴⁴.> 43. Mallamace, D., Fazio, E., Mallamace, F. & Corsaro, C. The Role of Hydrogen Bonding in the Folding/Unfolding Process of Hydrated Lysozyme: A Review of Recent NMR and FTIR Results. Int J Mol Sci 19, 3825, doi:10.3390/ijms19123825 (2018). 44. Wei, Y., Lee, D. K. & Ramamoorthy, A. Solid-state (¹³C) NMR chemical shift anisotropy tensors of polypeptides. Journal of the American Chemical Society 123, 6118-6126, doi:10.1021/ja010145l (2001).

REVIEWERS' COMMENTS:

Reviewer #1 (Remarks to the Author):

The authors have done a great job. I recommend the publication of the paper

Reviewer #2 (Remarks to the Author):

The authors have done a commendable amount of work for the revision. The rebuttal addressed many of my concerns. I'm very happy to see that the authors have clarified and included methodological details that should better allow others to try to reproduce the work. The text of the revised manuscript also seems much improved!

A minor concern, that has me in a somewhat head-spinning state of confusion. The authors have clarified that a major assay for the 'oligomers' is the A11 antibody. But a central finding of this paper is that the A11 antibody binds to the N21Q fibrils. Presumably this could be the case for other types of fibrils as well. Ever since Lansbury, there have been dozens or more learned reviews (and grant panels) saying that oligomers are more "biologically" important than fibrils. But here, the fibrils bind to the distinguishing reagent for the oligomers (the A11 antibody), and have toxicity properties usually attributed to oligomers. Are fibrils gonna make a comeback? Or perhaps both the A11 binding and toxicity have to do with structural features, that relegate the oligomer/fibril debate to an academic side-show. In any case, this paper presents some intriguing new data!